# The Matrix: Infinite-Horizon World Generation with Real-Time Moving Control

**Ruili Feng[1,5*‡] , Han Zhang[1,5*], Zhilei Shu[1,5*], Zhantao Yang[1,5*], Longxiang Tang[1,5*],**
**Zhicai Wang[1,5], Andy Zheng[3,5], Jie Xiao[1,5], Zhiheng Liu[1,5], Ruihang Chu[1],**
**Yukun Huang[2,5], Yu Liu[1†], Hongyang Zhang[3,4‡]**

[1]Tongyi Lab, [2]University of Hong Kong, [3]University of Waterloo, [4]Vector Insititute, [5]Matrix Team

[*]Equal Contribution, [†]Engineer Advisor, [‡]Project Leader

## Abstract

We present *The Matrix*, a foundational realistic world simulator capable of generating infinitely long 720p high-fidelity real-scene video streams with real-time, responsive control in both first- and third-person perspectives. Trained on limited data from video games like Forza Horizon 5 and Cyberpunk 2077, complemented by large-scale unsupervised footage from real-world settings like Tokyo streets, *The Matrix* allows users to traverse diverse terrains—deserts, grasslands, water bodies, and urban landscapes—in continuous, uncut hour-long sequences. With speeds of up to 16 FPS, the system supports real-time interactivity and demonstrates zero-shot generalization, translating virtual game environments to real-world contexts where collecting continuous movement data is often infeasible. For example, *The Matrix* can simulate a BMW X3 driving through an office setting—an environment present in neither gaming data nor real-world sources. This approach showcases the potential of game data to advance robust world models, bridging the gap between simulations and real-world applications in scenarios with limited data. See https://github.com/MatrixTeam-AI/matrix, https://matrixteam-ai.github.io/pages/TheMatrix/ for code data and project page.

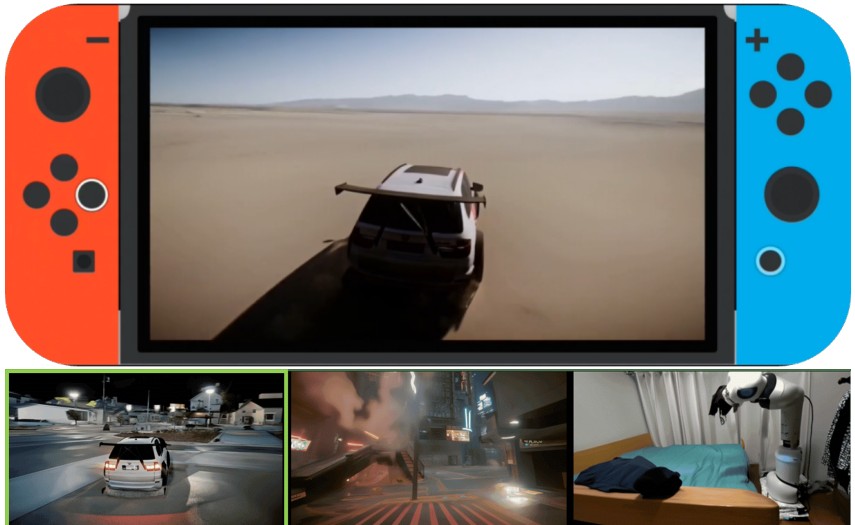

Figure 1: *The Matrix* is a foundational **realistic world simulator** capable of generating **infinitely long** 720p **high-fidelity real-scene** video streams with **real-time**, precise moving control. Due to size limitation, we recommend you to go to the website provided in the abstract for videos.

39th Conference on Neural Information Processing Systems (NeurIPS 2025).

# 1 Introduction

Neural-interactive simulation, a concept popularized by *The Matrix* (1999), envisions a world fully constructed by AI to replicate 20th-century human society. This paper takes an initial step toward realizing this vision by developing a world model that enables neural networks to 'dream' visually authentic environments. The result is an infinite-horizon, high-resolution (1280×720 pixels, 720p) simulation that supports real-time (16 Frame Per Second, FPS) interactive exploration across diverse landscapes, including deserts, grasslands, water terrains, and urban settings. Responding to real-time control signals, the world model predicts future frames in these environments in a streaming and auto-regressive fashion.

World models offer a promising solution to the overwhelming costs of video game development, which can easily run into tens or even hundreds of millions of dollars. Traditional game creation depends on engines such as Unity 3D, Unreal Engine, and Blender, each requiring substantial expertise, intensive asset preparation, and meticulous hyperparameter tuning. Furthermore, games built with these engines are often limited in reusability, as each new title demands a comprehensive redesign. In contrast, data-driven world models tackle these issues by minimizing the need for manual configuration, simplifying development workflows, and boosting scalability across projects.

Despite extensive research in world models [1], key challenges remain. First, prior studies have predominantly focused on simpler or less realistic video games, such as Atari [2, 3, 4], Mario [5], Minecraft [6, 7, 8], CS:GO [4], and DOOM [9], which fall short in replicating real-world fidelity. Second, current video generation techniques, like Sora [10], are constrained to short sequences of about 1 minute, forcing existing world models to assemble independently generated clips with noticeable transitions. Finally, achieving real-time generation remains a major hurdle. For example, state-of-the-art game generators such as Genie [11, 12] operate at speeds of 1 FPS, while Nvidia's Cosmos [13] is also incapable of generating real-time worlds. This paper addresses these limitations by introducing a scalable, high-fidelity world model in real time that enhances simulation realism and bridges the gap between virtual environments and reality. Notably, our world model is with strong domain generalization and real-time control. For example, our foundation model allows us to control BMW X3 driving through an indoor setting or in the sea—an environment present in neither gaming data nor real-world sources.

**Our Contributions.** Our contributions are as follows:

- We introduce *The Matrix*, a foundational simulator for realistic environments that generates infinitely long, high-fidelity 720p real-world video streams with real-time interactive control and strong domain generalization. Despite its capabilities, the model is lightweight, with only 2.7 billion parameters, and achieves a generation speed of 16 FPS on 8×A100 GPUs.

- At the core of *The Matrix* is a novel diffusion technique, the Shift-Window Denoising Process Model (Swin-DPM), enabling pre-trained DiT models [14] to extrapolate seamlessly for smooth, continuous, and infinitely extendable video creation. This technique holds potential for broader applications in long-form video generation.

- Additionally, we introduce *GameData*, a platform that autonomously captures paired in-game states—extracted from CPU memory—alongside corresponding video frames, significantly reducing labeling costs and complexity. This platform produces *Source*, a new training dataset for world models with action-frame paired data.

**Technical Advantages of *The Matrix*.** Our work advances the state-of-the-art in the following aspects: 1) Infinite Video Generation: *The Matrix* generates consistent, infinitely long videos using a streaming, auto-regressive approach. 2) High-Quality Rendering: *The Matrix* delivers realistic rendering at a resolution $1280 \times 720$. 3) Real-Time, Frame-Level Control: *The Matrix* operates in 16 FPS, providing real-time, frame-level control for interactive applications. 4) Domain Generalization: Trained with small amounts of supervised game data and large amounts of unsupervised internet videos, *The Matrix* achieves strong domain generalization to real-world settings.

While we do not claim *The Matrix* to be a foundational world model—since it is currently trained only on Forza Horizon 5, Cyberpunk 2077, and humanoid control tasks—we regard it as a proof of concept demonstrating that the aforementioned four aspects can be simultaneously achieved. With more diverse data, *The Matrix* has the potential to evolve into a generalized world model.

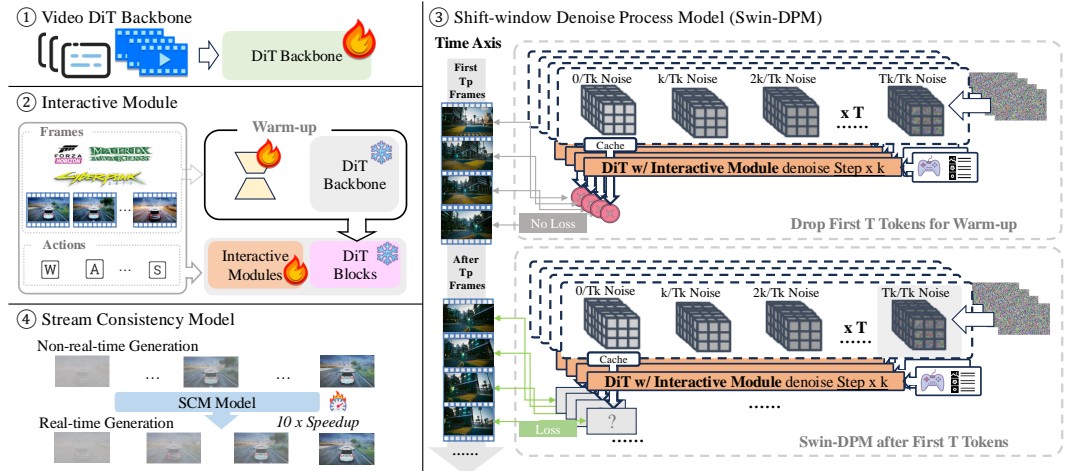

Figure 2: The training process of *The Matrix* begins with a pretrained video DiT backbone. First, the Interactive Module is warmed up using Synthesized Observations of Unreal Rendered Contextual Environments data with unsupervised LoRA to make subsequent training focus on movement, not visuals. Then, we train the Interactive Module for precise frame-level control. Swin-DPM enables infinite-length generation, and Stream Consistency Model is introduced to accelerate sampling to real-time speeds.

## 2   Related Work

**World Simulation.** Distinct from world models designed for agent learning, world simulation focuses on human interaction with neural networks through high-quality rendering, robust control, and strong domain generalization to real-world scenarios. This research explores two types of control: video-level and frame-level. In video-level control, a control signal is given at the start, and the model generates a responsive video sequence; notable examples include UniSim [15], Pandora [16], GameGen-X [17], MicroVGG [5], GAIA-1 [18], and Cosmos [13]. To approximate continuous control, this approach often stitches together independently generated clips, which may result in visible transitions. In contrast, frame-level control provides fine-grained adjustments every few frames, enabling more precise, responsive interactions similar to gameplay, as seen in examples like Genie [11], Genie-2 [12], DIAMOND [4], GameNGen [9], MineWorld [8], and Oasis [6]. Prior work in world simulation has typically focused on one of three aspects—video length, high resolution, or domain generalization—without addressing all three simultaneously. *The Matrix* uniquely stands out as a foundation model capable of generating infinitely long, high-quality videos with high resolution, frame-level real-time control, and strong generalization to real-world contexts.

## 3   Methods

Achieving granular control is notoriously challenging, as labeling actions at the frame level is typically cost-prohibitive. To address this, we develop the *GameData* platform, which autonomously captures paired data of in-game states (extracted directly from CPU memory) alongside corresponding video frames, significantly reducing labeling costs and complexity. Additionally, *The Matrix* incorporates an advanced Interactive Module that learns and generalizes game movement interactions from a limited amount of labeled data combined with extensive unlabeled data from both games and real-world environments. This enables *The Matrix* to deliver exceptional accuracy across diverse scenarios, while maintaining robust performance in the gaming domain.

Generating high-quality, real-time, and generalizable video simulations for infinite sequences presents additional technical challenges, often forcing previous simulators to compromise on one or more essential aspects. *The Matrix* overcomes these limitations by adapting the world model from a pre-trained video Diffusion Transformer (DiT) model [14], leveraging its extensive pre-existing knowledge and generation quality. To enable infinite-length generation, *The Matrix* introduces a novel diffusion approach, the Shift-Window Denoising Process Model (Swin-DPM), which allows

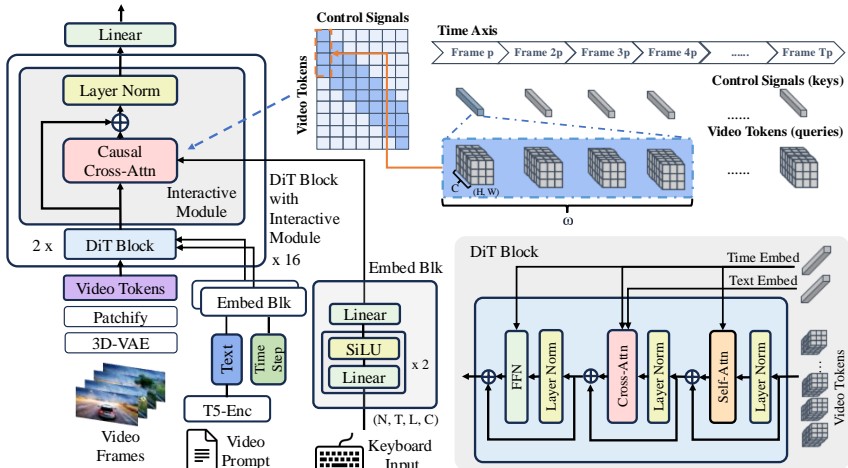

Figure 3: **The Interactive Module:** After every two DiT blocks, the module merges the keyboard inputs into the video token feature through a **Causal Cross-Attention Layer**, where each keyboard input is limited to influence only the current and subsequent $\omega$ tokens. Here, every $p$ frames are condensed into a single token.

the DiT model to extrapolate for smooth, continuous, and indefinitely long video creation. Finally, we fine-tune a Stream Consistency Model (SCM), accelerating inference to real-time.

**Video DiT Backbone.** As a preliminary, we introduce the video DiT backbone, adapted from the publicly available DiT models [19]. It employs a 3D Variational Auto-Encoder (VAE) to encode $T \times p$ video frames into $T$ video tokens. The backbone consists of 32 attention blocks, followed by a linear output head with LayerNorm [20]. Each attention block includes a self-attention layer operating on network features, a cross-attention layer linking conditions with self-attention outputs, and an FFN layer composed of two linear layers with a GELU activation [21] in between. See *Appendix Section A.1* for further details.

## 3.1 Model Components

*The Matrix* comprises three main components: a) an **Interactive Module** that interprets user intentions (e.g., keyboard inputs) and integrates them into video token generation; b) a **Shift-Window Denoising Process Model** (Swin-DPM) that enables infinite-length video generation; and c) a **Stream Consistency Model** (SCM) that accelerates sampling to achieve real-time performance. As shown in Figure 2, the model is fine-tuned from a pre-trained video DiT model through a three-stage process: first, we fix the DiT model parameters and train the Interactive Module; next, we train the Interactive Module and the DiT together following the Swin-DPM; finally, we optimize an SCM to accelerate inference to real time. The first two stages leverage both labeled gaming and unlabeled video data to enhance generalization, while the final SCM training focuses on labeled gaming data to reduce optimization complexity.

**Interactive Module.** The Interactive Module consists of an Embedding block (see Figure 3) and a cross-attention layer. Its primary function is to translate keyboard inputs into natural language that guides video generation. For example, pressing 'W' is interpreted as "The car is driving forward" in the *Forza Horizon 5* scenario, or as "The man is moving forward and looking up" when combined with an upward mouse movement in *Cyberpunk 2077*. For unlabeled real or game data, we apply a default description: "The camera is moving in an unknown way." To enhance robustness, we randomly replace labeled keyboard inputs with this default sentence during training with probability $q = 0.1$. To prepare for training, we first warmup the base DiT model for a few epochs using collected game and real-world data, fine-tuning a LoRA weight [22]. This process ensures that the Interactive Module focuses on learning interactions and movement patterns rather than simply fitting the video. Once translated, these natural language descriptions are processed by a T5 encoder [23] and transformed into a vector embedding through two linear layers and a SiLU layer [24] between them. This vector embedding is then concatenated with its corresponding video token and the next $\omega$ video tokens, where $\omega$ is a pre-defined causal relation range, typically set to $\omega = 4$, as is shown in Figure 3. We perform this cross-attention operation each time the DiT model completes an odd-numbered

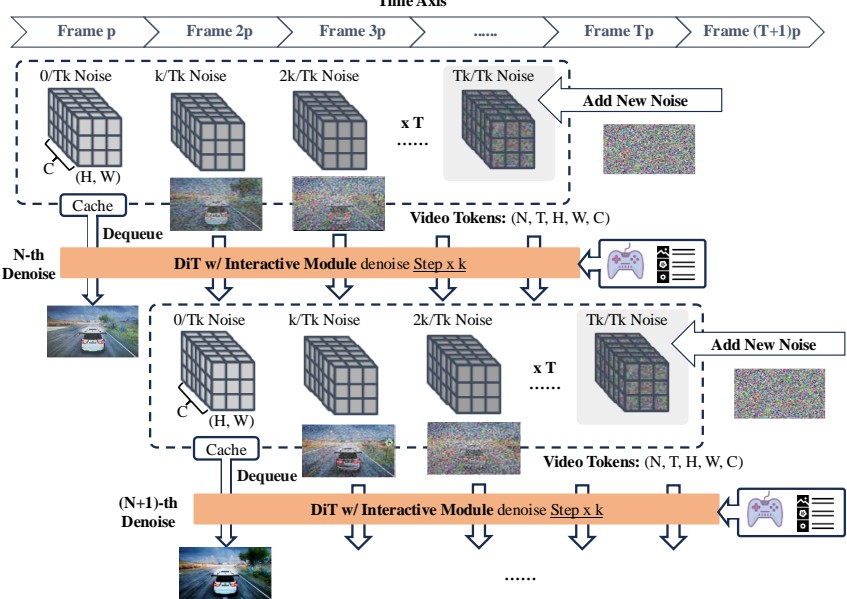

Figure 4: **Shift-Window Denoising Process Model:** The Swin-DPM transforms the traditional diffusion process into a streaming one, where $T$ video tokens with different noise levels are denoised simultaneously. After each token is fully denoised and dequeued for decoding, a new token of pure noise is added to the queue. The dequeued token is then copied to the cache, allowing it to continue participating in attention computations until the next token is dequeued.

self-attention step, enabling effective information exchange across frames and achieving precise, frame-level control for video generation.

**Shift-Window Denoising Process Model.** Typical DiT models are limited to generating only a few seconds of video, even when substantial spatial and temporal compression is applied via VAEs. This limitation is largely due to the high computational cost and memory demands of attention mechanisms over extended time durations. To address this, it becomes crucial to assume that temporal dependencies are confined within a limited time window, beyond which attention computations are unnecessary. Building on this idea, we propose the Shift-Window Denoising Process Model (Swin-DPM), which leverages a sliding temporal window to manage dependencies effectively and enables the generation of long or even infinite videos by producing tokens with a stride of $s = 1$. As is shown in Figure 4, within each window, a queue of video tokens undergoes denoising at various noise levels. After $k$ denoising steps (where $k \times T$ is the number of diffusion solver steps), the leftmost, lowest-noisy token is dequeued into a cache. To maintain the queue length, a new token with Gaussian noise will be then added to the rightmost position. Each cached token is re-appended to the window's token queue at noise level 0 until the next token is cached, allowing it to continue participating in denoising and ensuring continuity between different windows. The network of Swin-DPM is fine-tuned from a pre-trained DiT model. During training, we sample $2w$ video tokens, where $w$ is the window size. We usually set $w = T$. The first $w$ tokens are used solely for warming up Swin-DPM and do not participate in backpropagation; loss is computed only on the last $w$ tokens. At inference time, we follow the same setup: the first $w$ tokens are for warmup and are discarded, with the generated video starting from the $(w + 1)$-th token.

**Stream Consistency Model.** After extending the DiT model to Swin-DPM, we further address the need for achieving real-time rendering of the simulated world. A promising approach is to combine Swin-DPM with Consistency Models [25, 26], a leading method for accelerating diffusion. We use the Stream Consistency Model (SCM) [27], which distills the original diffusion process and its class-free guidance into a four-step consistency model while incorporating the denoising window design from Swin-DPM. The training procedure is illustrated in Figure 2. This integration results in a 10 - 20× acceleration in inference speed, reaching a rendering rate of 16 FPS.

### 3.2 Construction of the *Source* Dataset

To train *The Matrix* model, we construct the Synthesized Observations of Unreal Rendered Contextual Environments (*Source*) dataset, which consists of two components: synthetic game data from Unreal Engine and real-world, unlabeled footage. The synthetic game data, collected using the *GameData*

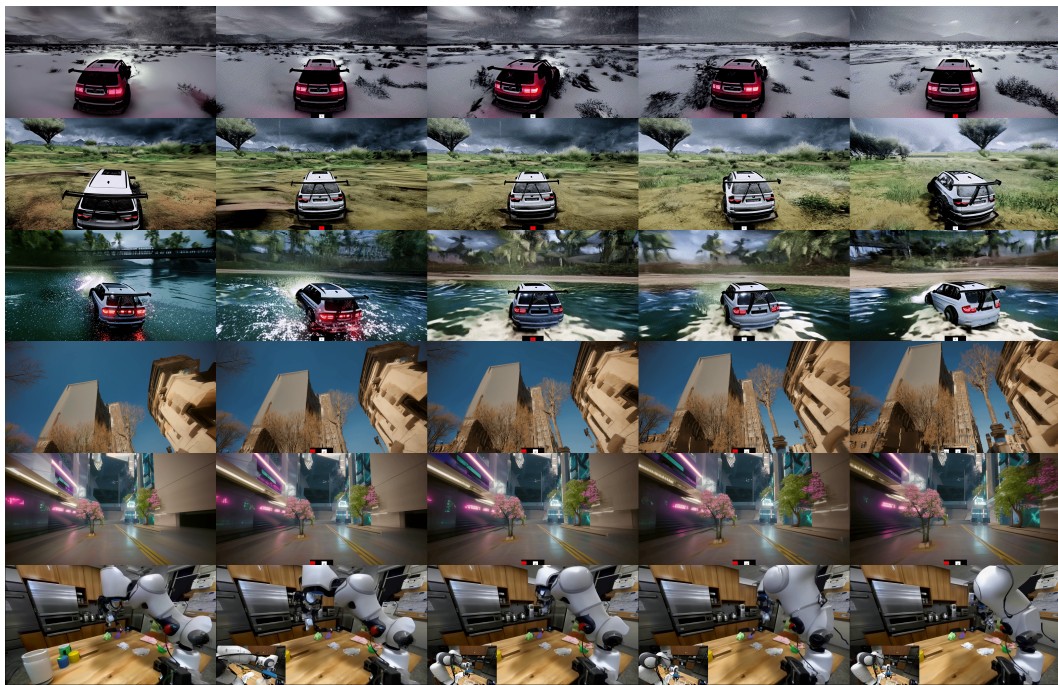

Figure 5: The results demonstrate frame-level precise control achieved by the Interactive Moduleacross diverse scenes, weather conditions, and movement modes. Due to size limitation, we recommend you to go to the website provided in the abstract for videos.

Platform, serves as supervised training data for precise motion control, while the real-world footage improves the model's visual quality and generalization to real-world scenarios.

After collection, the data is segmented into 6-second clips of continuous scenes and captioned using GPT-4o [28], resulting in a dataset of 750k labeled samples and 1.2 million unlabeled samples, all with 60 FPS. The labeled game data is further refined to ensure a balanced distribution of all possible game states. For more details on the dataset, see *Appendix Section B.2*.

**The *GameData* Platform.** The *GameData* Platform is built on open-source tools: *Cheat Engine* software [29], the *Reshade* plugin [30] for DirectX, and *OBS Recording* software [31]. *Cheat Engine* is used to capture in-game world status data, such as character $(x, y, z)$ positions and camera movements. This status data is aligned with recorded video frames to create per-frame action-video pairs and is also used to check if the character or camera is stuck and requires a reboot. We employ the *Reshade* plugin to remove all game UIs and HUDs and to standardize shading styles, providing a more consistent, low-complexity data source. Data for *Forza Horizon 5* is collected using autonomous scripts with random walking algorithms, while *Cyberpunk 2077* data is gathered manually with human operators running the *GameData* Platform. See *Appendix Section B.1* for more details on the *GameData* Platform.

## 4 Experiments

Since existing world models are not trained on *Forza Horizon 5* or *Cyberpunk 2077*, it is unfair to compare *The Matrix* against prior world models both qualitatively and quantitatively. Therefore, we focus on experiments only testing the properties of *The Matrix*. *The Matrix* is trained on 32x A100 GPUs in one week. All inference processes are conducted on 8x A100 GPUs.

**Training Details.** We train *The Matrix* on the *Source* dataset, using a pre-trained 2.3B parameter DiT model as the backbone, which generates 4 video tokens per second, each decoded into 4 frames by the VAE decoder [32]. To match this generation rate, we downsample the videos and keyboard inputs in the *Source* dataset accordingly. For all training cases, we first warm up the base DiT model on unlabeled *Source* data for 20,000 steps with a batch size of 32. Following this, we train the Interactive Module on labeled *Source* data for an additional 20,000 steps with the same batch size, introducing another 0.4B parameter. Next, we fine-tune *The Matrix* model using Swin-DPM over 60,000 steps, also with a batch size of 32. For the final Consistency Model distillation, we use the Swin-DPM

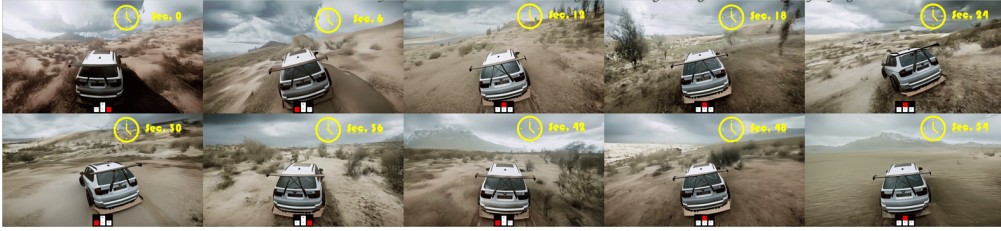

(a) Long 1-minute video generated by *The Matrix*.

Prompt: In a desolate *desert*, a white SUV is driving across the terrain. In an aerial shot, the vehicle is navigating through rugged landscapes, surrounded by parched vegetation and sparse trees. The camera follows the car's movement, capturing the tire tracks it leaves behind in the sand. In the distance, some buildings and mountains can be seen, while the sky is filled with clouds, with sunlight streaming through and casting rays of light.

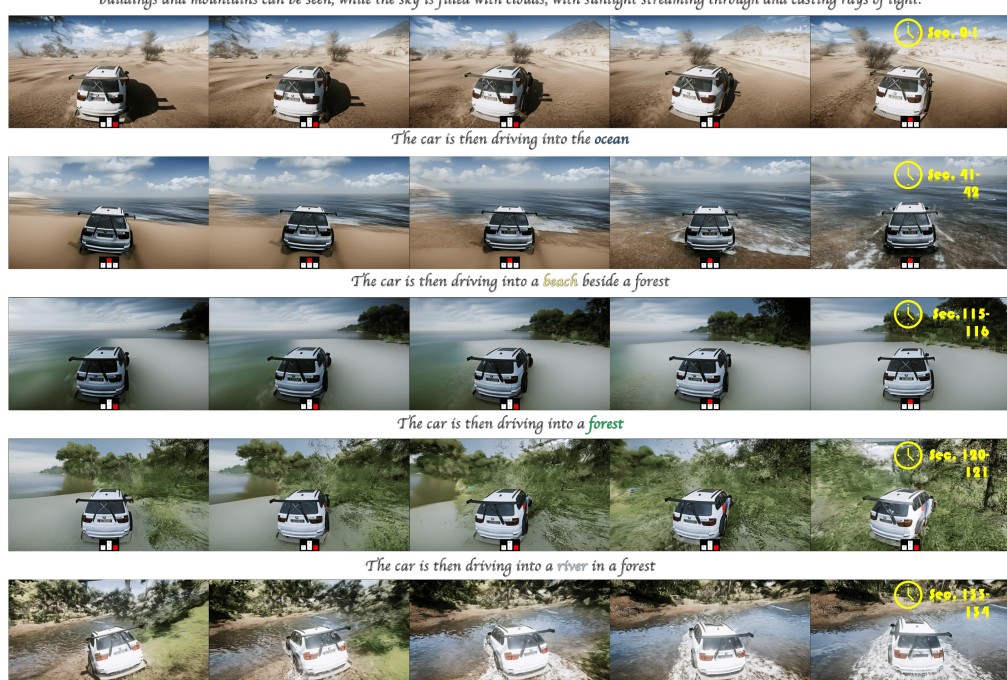

(b) A continuous 2.5-minute video generated by *The Matrix*, spanning multiple diverse scenes controlled through DiT text prompts.

Figure 6: Long world generation results by *The Matrix*.

checkpoints as a teacher model and train the student network for 10,000 steps with a batch size of 32. More details can be found in *Appendix Section A.2*.

**Metrics.** We evaluate performance using metrics for both general visual quality and movement control precision. For general visual quality, we use Fréchet Inception Distance (FID) [33], Fréchet Video Distance (FVD) [34], and CLIP Score [35] to assess text alignment. All metrics are evaluated on 2,048 seconds of randomly generated videos. To evaluate movement control precision, we generate 2,048 seconds of video based on keyboard inputs and text prompts from a fixed test set, then measure the Peak Signal-to-Noise Ratio (Move-PSNR) [36] and Learned Perceptual Image Patch Similarity (Move-LPIPS) [37] between the generated videos and real videos with ground truth movements.

## 4.1 Precise Frame-Level Interactions

In this section, we evaluate the effectiveness of the Interactive Module by testing its performance in three distinct scenarios: the *Forza Horizon 5* car driving scenario, the *Cyberpunk 2077* city walking scenario, and a robotic arm task from the *DROID* dataset [38]. We select 50,000 6-second clips from the *DROID* dataset, along with per-frame action labels of joint angles for seven joints, to form the training dataset. More details can be found in *Appendix Section B.3*. The third scenario is specifically designed to assess the effectiveness of *The Matrix* in embodied AI tasks. For all scenarios, we follow

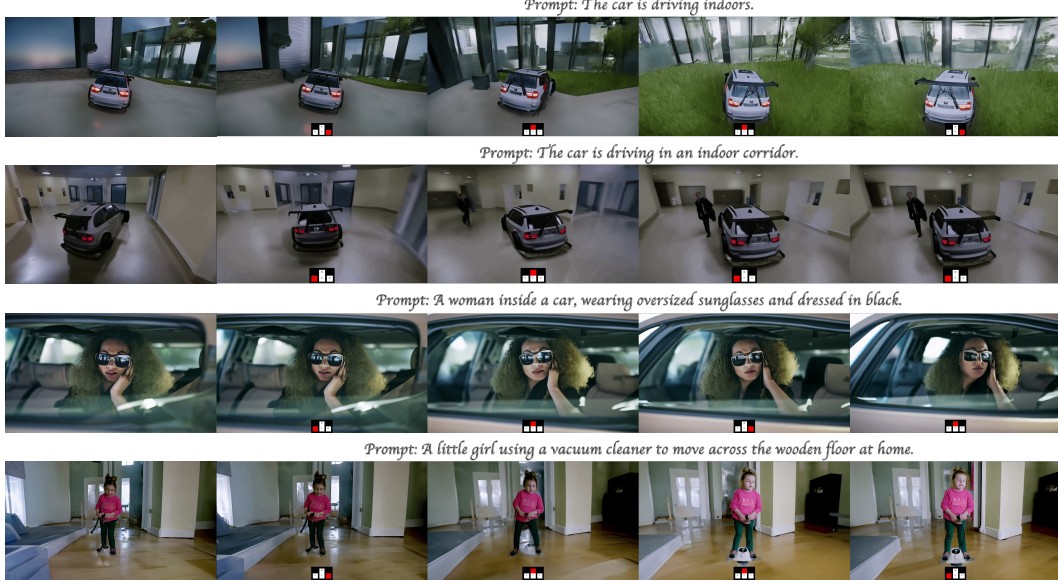

(a) *The Matrix* can generalize its precise movement control to unlabeled scenes and objects, such as driving indoors or making people move as instructed. Go to the website for videos

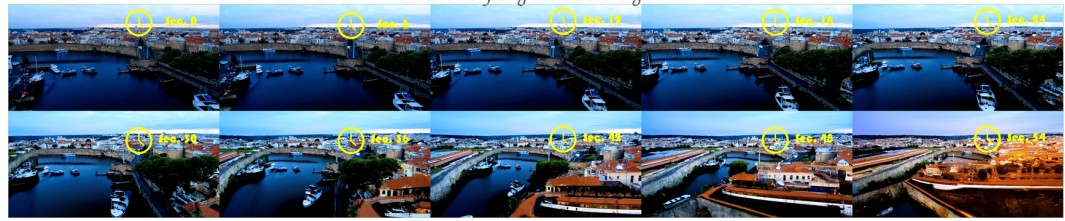

(b) *The Matrix* can also generate long, general videos by disabling the Interactive Module, acting as a powerful video generator.

Figure 7: Generalization ability of *The Matrix* on unseen scenes and objects.

the same training strategy: starting with a pre-trained DiT model, we first perform a warm-up using unlabeled data, followed by fine-tuning the Interactive Module with labeled data.

**Qualitative Results.** Figure 5 illustrates examples of *The Matrix*'s generated outputs across all scenarios. *The Matrix* demonstrates the ability to create vivid and dynamic worlds, accurately reflecting user interactions and intentions. It also models the physical behaviors within these environments, such as dust being kicked up when a car drives through a dry desert, or water splashing when it travels through a river. Additional examples of *The Matrix*'s generation capabilities are provided in *Appendix Section C.1*.

**Quantitative Results.** The last two columns of Table 1 present the quantitative evaluation of interaction precision, using LPIPS and PSNR metrics. The results demonstrate that Interactive Module significantly improves control precision, and this enhancement is maintained throughout the subsequent Swin-DPM and SCM processes.

### 4.2 Infinite-Horizon World Generation

Traditional world simulators focused on precise control often rely on small, auto-regressive generators trained from scratch to minimize the significant memory and time costs associated with pre-trained DiT models. However, this approach compromises visual quality and limits the full potential of world simulators. In this work, we introduce a world simulator leveraging pre-trained video diffusion models, enabling infinite-length world generation with real-time rendering capabilities. In this section, we present our evaluation of these advancements.

**Generating Infinitely Long Videos.** Figure 6 showcases examples of generating 1-minute long worlds across diverse scenarios, including desert, river, grassland, snow, and day-to-night transitions.

Table 1: Ablation study on the components of *The Matrix*. Note that there is a trade-off between inference speed, control precision, and rendering quality. **Move-LPIPS** and **Move-PSNR** are computed between the generated videos and test videos with ground truth movements.

| Component | Scene | Size | Speed | FVD↓ | FID↓ | CLIP↑ | Move-LPIPS↓ | Move-PSNR↑ |
|---|---|---|---|---|---|---|---|---|
| DiT Backbone | - | 2.3B | 1.41 FPS | 1016.30 | 318.10 | 0.30 | - | - |
| + Warmup | *Cyberpunk* | 2.3B | 1.88 FPS | 1429.45 | 183.24 | 0.28 | 0.125 | 27.80 |
| | *DROID* | 2.3B | 1.41 FPS | 1133.16 | 224.98 | 0.29 | 0.191 | 27.72 |
| | *Horizon 5* | 2.3B | 1.41 FPS | 1891.67 | 141.11 | 0.31 | 0.128 | 26.89 |
| + Interactive Module | *Cyberpunk* | 2.7B | 0.87 FPS | 1112.49 | 173.31 | 0.28 | 0.129 | 28.24 |
| | *DROID* | 2.7B | 0.87 FPS | 1200.82 | 237.66 | 0.30 | 0.180 | 27.90 |
| | *Horizon 5* | 2.7B | 0.87 FPS | 1211.30 | 119.20 | 0.27 | 0.125 | 28.98 |
| + Swin-DPM | *Horizon 5* | 2.7B | 0.8 FPS | 1651.50 | 163.27 | 0.24 | 0.113 | 29.90 |
| + SCM | *Horizon 5* | 2.7B | 16 FPS | 1936.79 | 153.80 | 0.23 | 0.109 | 29.73 |

During generation, we switch the DiT prompt to adapt the environment, as shown in Figure 6b. *The Matrix*'s capability extends beyond this; it can generate truly infinite-length videos, with additional 15-minute examples available in *Supplementary Videos*. Table 1 reports the video quality and control precision of *The Matrix* after training with Swin-DPM. While some visual quality is sacrificed, control precision remains strong, and the visual quality still surpasses previous world simulators.

**Real-Time Rendering.** We further investigate integrating SCM with *The Matrix*. As reported in Table 1, this integration highlights *The Matrix*'s real-time rendering capability, with a slight trade-off in visual quality and minimal loss in control precision, while significantly improving rendering speed from 0.8 FPS to 16 FPS.

### 4.3 Generalization to Out-of-Distribution Worlds

In addition to superior visual quality, a key advantage of using pre-trained video DiTs is their inherent ability to generalize across diverse scenes. We observe impressive generalization in *The Matrix*, showcasing the potential of future research into building world simulators with pre-trained DiTs.

**Generating Unseen Scenes.** With *The Matrix*, we can control a car in previously unseen scenes by describing the scenario in the prompt. The first two rows of Figure 7a demonstrate this capability, where the car is driven through indoor environments, which were not part of the *Source* dataset.

**Interacting with Unseen Objects.** A remarkable feature is *The Matrix*'s ability to generalize interaction with real-world objects. As shown in the last two rows of Figure 7a, by specifying a human as the center object in the prompt, we move the person in response to keyboard input.

**Generating Long Videos without Moving Control.** Though *The Matrix* is trained on the *Source* dataset, it can also function as a general long video generator. By disabling the Interactive Moduleand using only the DiT backbone trained after Swin-DPM, *The Matrix* can generate long videos corresponding to ordinary prompts. Figure 7b shows such an example, further proving *The Matrix*'s strength as a realistic world simulator.

## 5 Conclusion

We introduce *The Matrix*, a world simulator capable of generating infinitely long, high-fidelity video streams with precise real-time control. Trained on a blend of game data and real-world footage, *The Matrix* supports immersive exploration of dynamic environments, with zero-shot generalization to unseen scenarios. Operating at 16 FPS, it enables continuous, interactive simulations across diverse terrains, bridging the gap between virtual and real-world applications. This work highlights the potential of using game data to build robust world models with minimal supervision, and showcases the power of pre-trained video DiTs in enabling realistic, large-scale simulations.

**Limitations.** *The Matrix* requires 8 GPUs for deployment, making it resource-intensive. While it achieves a generation speed of 16 FPS, this is still slower than the originl frame rates. Moreover, the synthesized videos may occasionally exhibit physical implausibilities or temporal inconsistencies. Currently, *The Matrix* is trained on only two labeled games as a proof of concept. We leave the exploration of these limitations to future work. See more discussions in *Appendix Section E.1*.

**Broader Impacts.** We envision *The Matrix* as a foundational approach for AI-driven game generation, with the potential to transform the future of the gaming industry. The generated content blends real-world and synthetic game assets, enabling novel and immersive experiences.

## Acknowledgments

We would like to express our gratitude to the Alibaba Cloud Elastic Computing-Heterogeneous Hypervisor & Instance team for providing the Remote Gaming Console (RGC), which facilitates our data collection process. Special thanks to Bo Li, Min He, and their colleagues for their invaluable support and assistance. We also extend our appreciation to Microsoft and CD Projekt Red for developing their remarkable games, which played a key role in our data collection. A special acknowledgment goes to the Night City and Mr. V, who provided a peaceful environment for our research. We thank Shangwen Zhu from Shanghai Jiao Tong University, who contributed to the experiment design. We thank Deli Zhao from Alibaba DAMO Academy for discussions and consultant on the technique trend. We have special thank to Bryce Schmidtchen and Alberto Taiuti for valueable discussions. Finally, we thank Dr. Ping Luo of the University of Hong Kong for his insightful guidance on the collection of robotics data.

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

# A  More Related Works

**World Model for Agent Learning.** Developing world models for training agents has been a long-standing research focus, aimed at enhancing policy learning within simulated environments rather than solely achieving high-fidelity reconstructions of observations. This research involves two primary stages: 1) modeling the training environment by reconstructing observations, rewards, and continuation signals, often through a recurrent state-space model; and 2) utilizing this model to predict future states, enabling reinforcement learning to optimize robust policy functions. Studies indicate that this method provides sample efficiency gain of over 1000% compared to directly learning policies from real environments, shows resilience across diverse domains, and can outperform fine-tuned expert agents on a range of benchmarks and data budgets [7]. Key contributions in this area include Recurrent World Models [39], Dreamer (v1 [40], v2 [3], and v3 [7]), TD-MPC (v1 [41] and v2 [42]), DayDreamer [43], SafeDreamer [44], and MuDreamer [45]. Notably, MuZero [2] runs the self-play of Monte Carlo tree search to build world models for Atari, Go, chess and shogi, without external data.

# B  Details in Experiments

## B.1  DiT Backbone

The DiT backbone is adapted from the publicly available DiT models [19]. It consists of a patch embedding module, a caption embedding module, a timestep embedding module, 32 DiT blocks, followed by a linear output head with LayerNorm [20]. The followings provide details of each module within the DiT backbone.

**The Patch Embedding Module.** The patch embedding module employs a 3D convolution with a kernel size of $1 \times 2 \times 2$, followed by a reshape operation. Thus, the convolution can effectively process the video latent from the VAE encoder, and the reshape operation can further transform the feature into a sequence of tokens with 2,048 feature size. By using a 3D convolution, the module captures both spatial and temporal features, ensuring that the token sequence retains essential information from the video data.

**Caption Embedding Module.** The caption embedding module takes the caption token sequence encoded by the T5 model and further processes it through a two-layer FFN. Both the hidden feature size and the output feature size are set to 2,048, allowing the module to generate rich and high-dimensional representations of the caption data.

**Timestep Embedding Module.** The timestep embedding module is implemented as a sinusoidal embedding module followed by a two-layer FFN. Both the hidden feature size and the output feature size of this FFN are set to 2,048.

**DiT Block.** Each DiT block includes a self-attention layer operating on network features, a cross-attention layer linking conditions with self-attention outputs, and an FFN layer composed of two linear layers with a GELU activation [21] in between.

## B.2  Training Details

Upon obtaining the base DiT model, the training process consists of four distinct stages: (1) warm-up on unlabeled *Source*, (2) training of the Interactive Module, (3) fine-tuning using Swin-DPM, and (4) Stream Consistency Model distillation. Below, we first outline the common training configurations utilized across all stages, followed by a detailed description of each individual phase.

**Common Settings.** All training procedures were executed with an overall batch size of 32 and a learning rate of $1 \times 10^{-5}$. Mixed-precision training was employed using bfloat16 to enhance computational efficiency. During preprocessing, all video inputs were resized to a resolution of $1280 \times 720$ pixels and set to 16 FPS. For sequences exceeding 25,200 frames in length, we used the Deepspeed Ulysses sequence parallelism strategy [46], distributing the sequence across 8 GPUs to manage memory and computational demands effectively.

**Warm-Up on Unlabeled *Source* Dataset.** In the initial warm-up stage, we fine-tuned all linear layers of the base DiT model using Low-Rank Adaptation (LoRA) to tailor the model to the source data distribution [22]. The LoRA rank was set to 128, and the model was trained for 20,000 steps. This adaptation ensures that the model parameters are suitably adjusted to the characteristics of the unlabeled source dataset before advancing to subsequent training phases.

**Training of Interactive Module.** The second stage focuses on training the Interactive Module, each of which is integrated after every two consecutive DiT blocks, totaling 16 Interactive Module. During this phase, the parameters of the base DiT model were frozen to concentrate the training solely on the Interactive Module. This stage was conducted over 20,000 training steps, enabling the Interactive Module to effectively interface with the base model without altering its foundational parameters.

**Fine-Tuning Using Swin-DPM.** The third stage involves comprehensive fine-tuning of all model parameters, including both the base DiT model and the Interactive Module, utilizing the Swin-DPMapproach. This extensive fine-tuning was carried out over 60,000 steps, allowing for the refinement and optimization of the entire model to better capture data intricacies and enhance overall performance.

**Consistency Model Distillation.** In the final stage, consistency model distillation was performed using the model from the preceding fine-tuning phase as the teacher model. The student model was initialized with the teacher's weights to facilitate knowledge transfer. During distillation, we employed a one-stage guided distillation technique [47], incorporating Classifier-Free Guidance (CFG) into the student model. For the Ordinary Differential Equation (ODE) solver within the consistency distillation framework, we utilized the Euler solver with a single-step size of $25/1000$. This distillation process was conducted over 10,000 training steps.

## C The *Source* Dataset

### C.1 The *GameData* Platform

We build a framework, *GameData* Platform, for data collection. The framework consists of three components: Controlling, Simulation, and Observation.

**Controling.** In most games, we need to control a character to go to different scenes and make interactions. Intrinsically, the collected data can be reconstructed with initial states of game worlds and a series of control signal. In order to make the collected data clean and meaningful, instead of being stuck in one corner, we designed two different control systems, namely the automatic one and the manual one. For the automatic control system, we use Cheat Engine for pivotal data access, such as XYZ coordinates in games. These data can be used to determine whether the game has been stuck for some time. We detect the coordinates of a past period of time and determine whether they are covered in a circle of a given size. If the game is detected as stuck, we will reset the game state and restart the recording. Generally, the automatically generated control signals will move randomly, change direction, and change perspective. This is good enough for games that move on a 2D-like surface. However, for games moving in a 3D space, random signals will struggle with generating meaningful content, so we have to change to the manual system. Since our game is running and captured on cloud servers, human data collectors will observe the game through a low-definition streaming and control manually. Signals (from keyboards, mice, and gamepads) are translated and delivered through the socket server, and cloud servers will generate keyboard events through the virtual keyboard. Here, the latency between the control signals and the OBS screen recording is crucial. We eventually found that the control signals recorded on the cloud sever and the actual action responses in the recorded videos were generally no more than three frames apart. and in general, this delay is stable and can be subtracted directly from the timeline.

**Simulation.** The game runs directly on the cloud servers. we can directly copy the server images to get a large number of running instances. The recorded videos and control signals will be uploaded to the data center. We set up a series of video quality checks to filter out samples of low quality (still or overly noisy videos, and some undefined scenes). All games run at the highest quality while ensure the OBS screen recording does not get stuck. In order to avoid overly complicated situations, we

removed the NPCs and running vehicles in the game. We use the Reshade to adjust the game scenes to make it more reality-like.

**Observation.** We use OBS as the screen recorder. One can use scripts to control OBS for automatic recording. We recorded the game at native resolution of $2560 \times 1600$ (higher resolutions may cause the game and recording to lag). For the reality of the recorded videos, we removed GUIs and texts in the game through a Reshade plugin, namely ReshaderEffectShaderToggler.[1] It can turn off the rendering of GUI related shaders in the game while left the native video untouched.

**Forza Horizon 5.** In Forza Horizon 5, a telemetry mechanism can be used for game status retrieving. We can access the real-time game data through socket after checking on the telemetry option in settings. An example script for data listening can be found here.[2] We can access XYZ coordinates, velocities and accelerations. We use these data for stuck detection and sample filtering. Since Forza Horizon 5 is a game that mainly takes on 2D area, we apply automatic pipeline that randomly walking on different game scenes (like dessert, grassland, the watery and the snowy areas). Control signals are simplified to going forward, turning left and turning right. During the data collection stage, if XYZ of is still for several seconds, the controller will try to move back. And if the $40$ position points collected during the last $40$ seconds can be covered with a circle with radius of $80$ meters, the controller will try to teleport the car to a random position. After raw sample collection, we apply some strategies to filter out samples of low quality. We use the acceleration data to detect if the car has collided with anything, and drop these video clips with collision. Sometimes the car is moving backward while the controlling input is moving forward, this is because the direction of movement in the game is to provide acceleration. We filter out data with a large angle between acceleration and velocity. Due to some problems in the game itself, the video often changes suddenly at some time. We filter out video clips with large average error between any two adjacent frames.

**Cyberpunk 2077.** Cyberpunk 2077 is a game that offers realistic visuals and lighting effects. Due to the complexity of the game terrain, we have to choose the manual pipeline. For simplicity, the actions in game are reduced to two separated inputs. The first one makes the character move forward or stop. And the second one makes the direction of the character's sight move up, down, left and right. We disable the NPCs and moving vehicles with game mod. During the data collection, players observe the game through low-definition OBS streaming and send control signals. The signals are then mapped into "W" (moving forward) / "U" "D" "L" "R" (up, down, left and right) on the cloud servers. We access and record the XYZ coordinates of player through Cheat Engine. These coordinate sequences are then used for filtering out video clips where collisions occur between the character and the game scene.

## C.2 The *Source* Dataset

We present the *Source* dataset from three perspectives: basic information, the annotation method used to convert the original data from *GameData* **Platform** to our desired format, and the filtering method applied to remove undesirable data.

### C.2.1 Basic Information

The *Source* comprises data from both Forza Horizon 5 and Cyberpunk 2077. For Forza Horizon 5, we collected approximately 1,200,000 pairs of video and control signals, while for Cyberpunk 2077, we gathered around 1,000,000 such pairs. All collected videos have a duration of approximately 6 seconds, recorded at 60 FPS. For Forza Horizon 5, we specifically collected data from multiple scenes, including deserts, oceans, water bodies, grasslands, and fields. The videos from different scenes are illustrated in Figure A1, along with the distribution of data volume for each scene Figure A2 (a). For Cyberpunk 2077, we focused on gathering data from urban environments that feature a significant number of tall buildings.

In Forza Horizon 5, the dataset includes only three distinct control signals: "moving forward" (denoted by "D"), "moving forward and turning left" (denoted by "DL"), and "moving forward and turning right" (denoted by "DR"). In contrast, the data for Cyberpunk 2077 encompasses five different

---

[1]https://github.com/4lex4nder/ReshadeEffectShaderToggler
[2]https://github.com/jasperan/forza-horizon-5-telemetry-listener

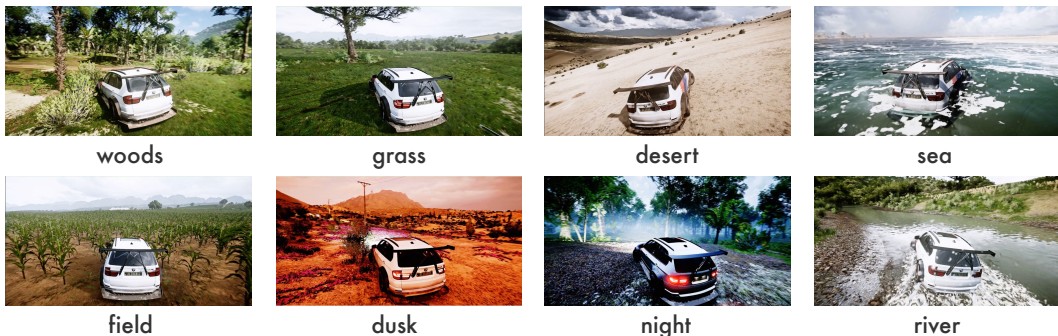

woods      grass      desert      sea

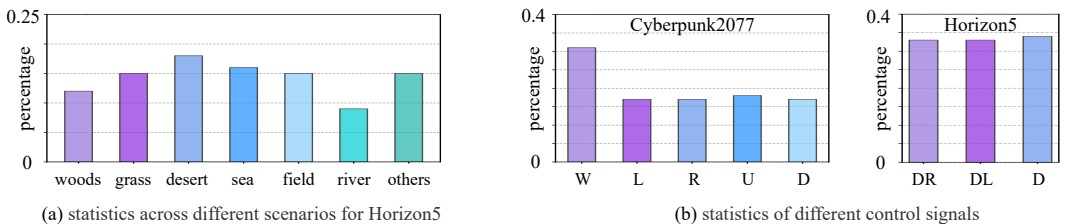

field      dusk      night      river

Figure A1: Examples of Horizon5 across different scenarios.

(a) statistics across different scenarios for Horizon5

(b) statistics of different control signals

Figure A2: (a) The statistics for Horizon5 across different scenarios include woods, grass, desert, sea, fields, rivers, and others. The results indicate that the quantity of data across these scenarios is relatively balanced. (b) The statistics of different control signals for Cyberpunk 2077 and Horizon5. In Cyberpunk 2077, the percentage of the "move forward" signal is relatively high, while other steering control signals are distributed more evenly. In Horizon5, all three control signals are evenly distributed.

control signals: "moving forward" (denoted by "W"), "turning left" (denoted by "L"), "turning right" (denoted by "R"), "looking upward" (denoted by "U"), and "looking downward" (denoted by "D").

### C.2.2    Annotation Methods

The original data from *GameData* **Platform** typically has a duration of around 10 minutes, which is excessively long for training *The Matrix*. Therefore, we use FFmpeg [48] to segment these videos into 6-second clips. Next, we extract the corresponding control signals from the complete set of signals. We then use InternVL [49] to generate captions based on 12 uniformly extracted key frames from the videos. After the captioning process with InternVL, we perform manual corrections on the generated captions to eliminate obvious errors.

### C.2.3    Filtering Methods

After the annotation step, a significant amount of undesired data remains, which could disrupt the training of *The Matrix*. To address this, we employ five filtering methods to eliminate these problematic data points, which we introduce as follows. Note that for Cyberpunk 2077, since we utilize human data collection rather than automatic methods, many of the following issues do not exist.

**Balance Control Signals.** Balancing the number of different control signals is beneficial for the training of *The Matrix*. The process of balancing control signals consists of three steps: 1) First, we analyze the distribution of control signals for each 6-second video and record the results. 2) Next, we assess the overall distribution of control signals across the entire dataset to identify the most frequently occurring control signal. 3) Finally, we remove some data points that contain the highest proportion of this predominant control signal. We repeatedly implement the second and third steps until the distribution is relatively balanced. The distribution results for Forza Horizon 5 and Cyberpunk 2077 are reported in Figure A2 (b). We provide the pseudocode for the algorithm in Algorithm 1.

**Algorithm 1** Control Signal Balancing Algorithm

---

**Require:** Dataset $D$ containing control signals from 6-second videos
**Ensure:** Balanced Dataset $B$
1: Initialize $B$ as an empty set
2: **for** each video $v$ in $D$ **do**
3:     Analyze the distribution of control signals in $v$
4:     Record the results for $v$
5: **end for**
6: **while** not isBalanced($B$) **do**
7:     overallDistribution $\leftarrow$ Assess the overall distribution of control signals in $D$
8:     mostFrequentSignal $\leftarrow$ Identify the most frequently occurring control signal from overallDistribution
9:     $D \leftarrow$ Remove data points from $D$ that contain mostFrequentSignal
10: **end while**
11: Set $B \leftarrow D$
12: **return** $B$

---

**Detect and Remove the Data with Collisions.** In Forza Horizon 5, randomly generated control signals often cause the car to collide with walls or rocks. Additionally, the car may be struck by other vehicles. These collisions can severely disrupt the training process, making it essential to identify and remove collision-affected data. Our analysis revealed that collisions consistently result in abrupt changes in acceleration over a very short time. Thus, we use significant variations in acceleration as a reliable indicator of collision events and discard any corresponding data to maintain the integrity of the training process.

**Detect and Remove Stuck Data.** In Forza Horizon 5, after colliding with walls or rocks, the car often gets stuck; even when the "D" key is pressed, the car fails to move. This stuck situation complicates the training data and negatively impacts the performance of *The Matrix*. Therefore, we need to detect and remove such instances. Detecting when the car is stuck is relatively straightforward—we simply calculate the distance the car has traveled within the video. If this distance falls below a certain threshold, we conclude that the car is stuck and discard the corresponding data.

**Detect and Remove the Data with Mismatched Motion and Control.** As introduced in appendix C.1, to quickly resolve a stuck situation, the car will move backward when stuck. As a result, it is possible for the car to still move backward at a slower speed even when the "D," "DL," or "DR" keys are pressed. Similar situations may arise when "DL/DR" is pressed for a long period and then switched to "DR/DL." Although the acceleration is directed to the right/left, the car may continue to move in the opposite direction for a brief period. We refer to this as mismatched motion and control, which complicates the training process. To address this issue, we calculate the directions of both the acceleration and the car's movement. If the angle between these two directions is too large, we discard the corresponding data.

**Detect and Remove Artifacts.** In Forza Horizon 5, visual artifacts can occur when a car collides with obstacles like trees, introducing distortions into the generated videos. To filter out such corrupted data, we detect variations in pixel values across consecutive frames. Our analysis shows that applying a high threshold effectively identifies all videos containing these artifacts, enabling their removal.

### C.3 The *DROID* dataset

#### C.3.1 Basic Information

DROID is a large, diverse robot manipulation dataset containing 76k demonstration trajectories (350 hours of interaction) collected across 564 scenes and 86 tasks over 12 months by 50 collectors worldwide. It aims to improve the performance, robustness, and generalization of robotic manipulation policies. DROID uses the same hardware setup across all 13 institutions to streamline data collection while maximizing portability and flexibility. The setup consists of a Franka Panda 7DoF robot arm, two adjustable Zed 2 stereo cameras, a wristmounted Zed Mini stereo camera, and an Oculus Quest 2

headset with controllers for teleoperation. Everything is mounted on a portable, height-adjustable desk for quick scene changes.

### C.3.2 Filtering Methods

**Remove Overly Complex Scenes and Balance Different Scenes.** DROID is a collaborative effort involving multiple laboratories and contains data from 11 different environments, including domestic scenes like kitchens and bedrooms, as well as industrial settings such as factories and laboratories. Due to the complexity of these scenes, which poses challenges for subsequent captioning and video learning, we first classify the data based on scene labels. For each category, we manually select 50 less complex scenes. We then use DINO to encode and extract semantic features to calculate the mean, removing outliers within each scene based on this semantic mean. To balance the number of training samples across scenes, we ensure that the final dataset contains an approximately equal number of samples from each scene after outlier removal.

**Filtering Frames Without Arm Presence and Removing Failed Executions.** Since some frames in the videos do not contain the robotic arm or have only a small visible area, we use Grounding DINOv2 [50] to remove such frames. If more than 20% of the frames in a video meet this condition, the entire video is discarded. Additionally, to ensure accuracy in control, we remove data where the robotic arm fails to follow the intended trajectory successfully. Finally, we use the spatial position of the robot gripper as a condition for each frame in the training.

## D  More Examples

### D.1  Generalization

We provide more results on the generalization ability of *The Matrix* in Figure A3.

### D.2  Long Video Generation

We provide several long video generation demos in the *Supplementary Video files.* Please check them after Unzip. **All videos are heavily compressed to satisfy the supplementary file size limit.**

## E  Discussions

### E.1  Limitations

While The Matrix shows strong potential, there are several areas that remain challenging and should be the focus of future research and development:

- Physical Implausibilities and Temporal Inconsistencies: While the model generates visually impressive videos, there are occasional physical implausibilities and temporal inconsistencies. These aspects highlight opportunities for enhancing realism in future iterations.
- Limited Training Scope: Currently, The Matrix has been trained on only two labeled games (Forza Horizon and BMW). While these initial results show great promise, broadening the model's training across a wider variety of games and environments will be crucial for improving generalization and real-world transferability.
- Handling of Dynamic Objects: The Matrix performs well with static objects, such as the BMW car, but handling dynamic objects that exhibit complex behaviors remains an ongoing challenge. Addressing this limitation will be essential for expanding the model's capabilities in more dynamic, real-world scenarios.

These limitations reflect important areas for future study and refinement. While The Matrix is a strong proof of concept, overcoming these challenges will enhance its applicability and performance in diverse environments and more complex scenarios.

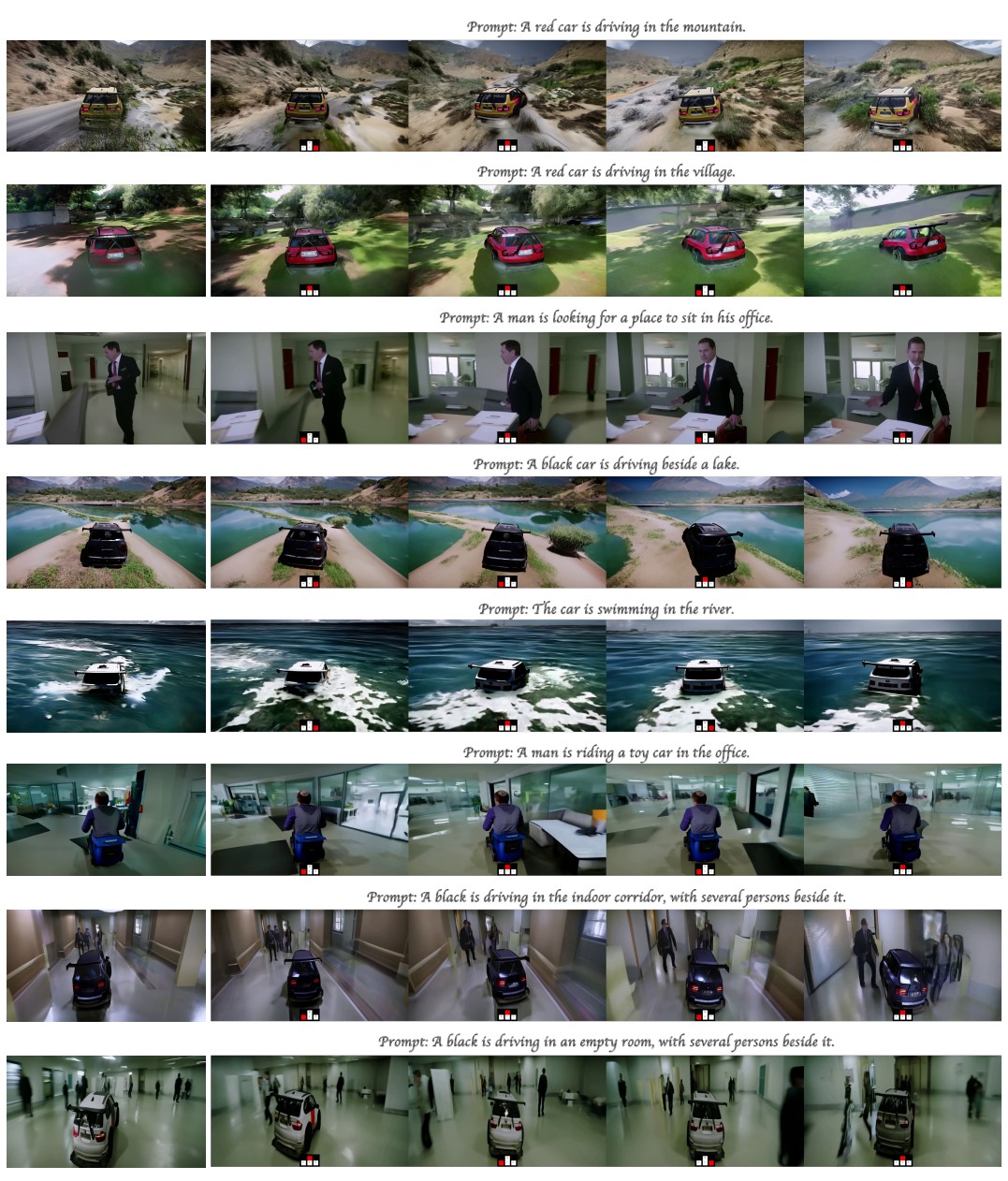

Figure A3: More generalization results of *The Matrix* on unseen scenes and objects.

