# OpenReview forum: "The Matrix: Infinite-Horizon World Generation with Real-Time Moving Control"
_NeurIPS.cc/2025/Conference — NeurIPS 2025 poster_

### Official Review · Reviewer_6pVJ · 2025-06-17

**Clarity:** 3
**Significance:** 3
**Originality:** 2
**Rating:** 4
**Confidence:** 4

**Summary:**

The work proposes an interactive video generator capable of: generating videos in real-time, generate long and high-resolution videos, condition generation on actions specified on a frame-by-frame basis, transfer knowledge learned from synthetic video game environments to real-world scenarios.
To realize such model, the authors consider a pretrained DiT video generator, introduce "Interactive Module" layers to a accept the control signals, Shift-Window Denoising Process Model to generate long videos auto regressively, and Stream Consistency Model to accelerate inference and achieve real-time generation.
Some quantitative ablations and qualitative evaluations are shown to support the paper claims.

**Questions:**

The paper is explained clearly and my doubts regarding the work stem could be clarified with more evaluation. Several technical design choices or components of the training recipes as highlighted in previous review points do not find an ablation. I would reserve a higher score for the work if it presented experiments evaluating:
- Data mixture: How important it is to use real data? How much data is needed? Are multiple videogame data needed to achieve real-world controllability?
- Action representation: Is it necessary to translate keyboard and mouse actions to textual descriptions?
- Swin-DPM and SCM: What is the effect of their main hyper parameters on final quality of the model?
- How each component affects real-world transfer performance: present metrics that link each technical contribution to the amount of real-world transfer performance they are responsible for

**Ethical Concerns:**

["NO or VERY MINOR ethics concerns only"]

**Final Justification:**

During their rebuttal the authors addressed my outstanding concerns through additional experiments, so I update my score to lean towards acceptance.

**Limitations:**

The paper presents a limitation section, but does not describe method limitations deeply. The section should mention the issues reported in the review regarding lack of memory, limited environment understanding, very limited ability to perform real-world knowledge transfer and difficulties in handling dynamic objects besides the BMW car.

**Paper Formatting Concerns:**

- LL46 states the model constitutes a foundational simulator, but LL65 says there is no claim that the work is a foundational world model.
- LL 91 claims "exceptional accuracy across diverse scenarios"
- LL 243 claims "impressive generalization"
- The limitation section should include issues reported in the review
I suggest revising such points to represent more fairly the work and its contributions

- LL 221 typo

**Quality:**

2

**Strengths And Weaknesses:**

QUALITY
- The generated driving results are compelling, showing 16fps generation of high resolution videos with fine grained control for lengths up to 15 minutes. This is a difficult result to achieve. Driving results however show limitations in the situations the model can simulate. The model does not seem able to handle properly jumps, with the car appearing to always drive on mostly flat surfaces despite the presence of dunes in the desert environment. The model has little memory for previously generated part of the environment. When the car turns back to a region observed few seconds ago, a new scene is generated instead.
- I appreciate the attempt to transfer the learned interaction to real world scenes, but quality of results in this task is limited and not accompanied by evaluation. Non-car object such as people do not appear to be well controllable and car driving results in unseen environments have rather limited quality. The relationship between each proposed framework components and real-world transfer performance is not shown.
- Evaluating interactive video generators is challenging. For this reason the paper contains no comparisons to previous baselines. The ablations section however does not present such challenges and is not very extensive. The reader could have been provided with better insights into the operated choices for the Shift-Window Denoising Process Model and Stream Consistency Model hyper parameters, more designs for the Interactive Module (To cite one, can we use discrete actions rather than converting them to text?) and deeper ablations into data (To cite one, how much synthetic data is needed? What is the impact of jointly training with real data? How much does using different video games help real-world transfer?). The significance of the paper is diminished by the lack of deeper quantitative analysis.


CLARITY
The paper describes the proposed system and data collection pipeline with good clarity.


SIGNIFICANCE
- The paper positions itself under the category of works aiming to produce interactive world simulators. This is a growing area with several previous works as "GameGAN", "Playable Video Generation", "Genie", "DIAMOND, "GameNGen", "Oasis" and many concurrent works such as "Matrix-Game", "WorldMem", "DeepVerse". This area of work is likely to become of broader interest in the context of large scale video generation models.
- The paper presents a recipe for creating an interactive video generation model starting from a pretrained one which can be of interest for the community
- The paper presents a meaningful procedure for data collection that can be re-used by the community to perform data collection on synthetic video game environments
- The learned actions focus mostly on camera manipulation (moving the car, egocentric view or the subject). The work focuses on simpler scenarios, e.g. by removing pedestrians and cars when acquiring Cyberpunk 2077 data. This poses a limit on the capabilities the model can develop and reduces significance. This is a challenging aspect across multiple concurrent works.

ORIGINALITY
- Originality of the paper is fair. The problem of building interactive video generator is established. The employed techniques for action injection consist in a cross attention layer with causality, Shift-Window Denoising Process Model adopts the established idea (see "Diffusion Forcing") of adopting progressively decreasing time steps per successive frames or blocks of frames, Consistency Models have been employed widely to speedup video generators.
- The most original component of the paper lies in the attempt of generalizing results to real-world videos. Quality of results is limited in this task and evaluation not present.

---

> ### Author Rebuttal · Authors · 2025-07-31
>
> **Dear Reviewer 6pVJ:**
>
> Thank you for your insightful feedback! We are encouraged by your appreciation of the significance and quality of our work. Below, we address your concerns in detail.
>
> ---
>
> ### **Concern 1: Swin-DPM is similar to Diffusion-Forcing**
>
> Thank you for raising this point! While Diffusion-Forcing is indeed an impressive approach, there are notable differences between it and our method, Swin-DPM:
>
> - **Length of Generated Videos:** Diffusion-Forcing struggles with the accumulation error of auto-regressive models, making it challenging to generate videos longer than 10 minutes. In contrast, Swin-DPM can generate stable-quality videos over extended periods, such as half an hour, far beyond the capabilities of Diffusion-Forcing.
>
> - **Real-Time Capabilities:** Diffusion-Forcing is not designed for real-time applications and does not focus on optimizing latency, video frame production, and consistency within limited memory, computational resources, and time constraints. Swin-DPM, on the other hand, incorporates techniques to significantly reduce latency and improve real-time video generation.
>
> ### **Concern 2: Importance of Data Mixture**
>
> We agree that data mixture is important for enhancing the generalization ability of the control model. However, we believe that multiple game datasets are not strictly necessary to achieve this. Our approach shows that a well-designed dataset and careful training can lead to strong generalization capabilities, even with a single game dataset. Below we report the results of two different settings, trained with game data only and trained with real-world data fusing. The results demonstrate that training fusing real-world data can improve generalization on real-world scenes.
>
> **Table: Performance Comparison of Training Strategies**
>
> | Training Strategy           | Game Scene Moving-PSNR ↑ | Game Scene Moving-LPIPS ↓ | Real-World Scene Moving-PSNR ↑ | Real-World Scene Moving-LPIPS ↓ |
> |----------------------------|-------------------|--------------------|-------------------------|--------------------------|
> | Game Data Only             | 29.92             | 0.112              | 26.14                   | 0.199                    |
> | Fusing with Real-World Data| 29.90             | 0.113              | 27.79                   | 0.172                    |
>
> In addition, we report the results of varying the rate at which the action label is replaced with NULL, where the User-Study Accuracy is based on a manual inspection of the generated videos, where we counted how many frames we considered correctly aligned with the control inputs. Please note that these results were reproduced from training on a small subset of the full dataset and may not align with the results presented in the paper.
>
> **Table: Effect of Replace Rate on Evaluation Metrics**
>
> | Replace Rate      | FVD ↓ | Moving-PSNR ↑ | Moving-LPIPS ↓ | User Study ACC ↑ |
> |-------------------|-------|---------------|----------------|------------------|
> | 0.00              | 1344  | 10.42         | 0.600          | 0.89             |
> | 0.05              | 1314  | 10.45         | 0.594          | 0.87             |
> | 0.10              | 1311  | 10.44         | 0.594          | 0.88             |
> | 0.15              | 1318  | 10.44         | 0.594          | 0.86             |
>
> ### **Concern 3: Action Representation**
>
> For the main task, it is not strictly necessary to transfer keyboard inputs into text representations. However, we chose to do so primarily for generalization reasons. The generalization of control in out-of-distribution (OOD) scenarios is achieved by altering the text prompts. The model is pretrained using the DiT model and inherits its understanding of text-based commands. By randomly dropping control signals during training, the model learns to manipulate the movements of real-world objects.
>
> We hope to expand this in future work by collecting a broader range of interaction types and using language as a unified protocol to understand and merge them into the model. This will enable the model to generate novel interactions independently. We did not implement this idea in the current work due to the substantial workload involved, but we see it as an important avenue for future research. This future direction is also the reason we converted action labels to text.
>
> We further conducted an experiment to validate the effectiveness of using transfer action labels in the form of one-hot labels instead of text prompts. Due to time constraints during the rebuttal period, the model may not have been fully trained (note that the numerical results may not be aligned with the paper), but the preliminary results suggest that text prompt-style actions outperform one-hot labels across all metrics. Specifically, here User-Study Acc. suggests we manually look into generated videos and count how many percentage of frames we think is correctly aligned with controls.
>
> We hypothesize the following reasons for this result:
>
> - **Text Prompts Provide Semantically Rich Representations:** Text prompts offer distinct and semantically meaningful representations for actions. In contrast, one-hot labels are less expressive. For example, concepts like "press W for 1 second, then press S for 1 second" or "moving irregularly" are difficult to capture with one-hot labels but can be easily represented with text-based prompts.
>
> - **Base DiT Model Familiarity:** The base DiT model has been pretrained with text prompt embeddings, and it cooperates well with these embeddings. Since the model is already familiar with text-based representations, it is more effective when using them compared to one-hot labels.
>
> **Table: Comparison of Action Spaces on Evaluation Metrics**
>
> | Action Space | FVD ↓  | Moving-PSNR ↑ | Moving-LPIPS ↓ | User Study ACC ↑ |
> |--------------|--------|---------------|----------------|------------------|
> | one-hot      | 1311   | 10.42         | 0.590          | 0.72             |
> | text         | 1288   | 10.52         | 0.600           | 0.90             |
>
>
> ### **Concern 4: Swin-DPM and SCM Parameter Effectiveness**
>
> Regarding Swin-DPM, the main parameter we focus on is the stride size. We have evaluated the effect of this parameter through ablation experiments, which we have detailed below. For SCM, there are relatively few parameters to tune. It uses a default setting of 4 diffusion steps with no conditional guidance (CFG). Could you please clarify which specific parameters you are referring to? This would help us provide more targeted insights on the parameter effectiveness.
>
> **Table: Effect of different strides over VBench Metrics**
>
> | VBench Metrics         | Stride 1 | Stride 2 | Stride 3 | Stride 4 |
> |------------------------|----------|----------|----------|----------|
> | subject_consistency    | 0.862    | 0.873    | 0.882    | 0.885    |
> | background_consistency | 0.900    | 0.911    | 0.914    | 0.917    |
> | imaging_quality        | 0.662    | 0.675    | 0.689    | 0.695    |
> | motion_smoothness      | 0.976    | 0.976    | 0.978    | 0.977    |
> | dynamic_degree         | 0.99     | 1.00     | 1.00     | 1.00     |
> | aesthetic_quality      | 0.527    | 0.533    | 0.530    | 0.528    |
>
> ### **Concern 5: How Each Component Affects the Performance of Real-World Transfer**
>
> We recognize that real-world transfer is an important aspect of our work, and each component of our method contributes in different ways to improving its effectiveness in this regard. We are currently working on additional experiments to further analyze the influence of each component on the real-world transferability. We will include these findings in the revised version of the paper and the rebuttal as soon as possible to provide a more detailed breakdown of how each component affects the performance.
>
> ---
>
> We hope these clarifications address your concerns. Thank you again for your thoughtful feedback and for considering our work.

---

> > ### Comment · Reviewer_6pVJ · 2025-08-07
> >
> > I thank the authors for finalizing their rebuttal results.
> >
> > The additional results provided for real world transfer performance and promise to revise the paper's limitations address my remaining concerns.
> >
> > I will thus update my score accordingly.

---

> ### Comment · Reviewer_6pVJ · 2025-08-03
>
> Dear authors, thank you for the comprehensive rebuttal.
> The rebuttal addresses a part of my concerns:
> - The discussion on the use of textual rather than discrete actions is convincing
> - The showcased ablations on data mixture (mixing with real-world data), stride parameter and action representation provide more insights into the work
>
> In their rebuttal, the authors promise additional results on real-world transferability which I'm looking forward to see in this rebuttal period to finalize my recommendation.
>
> I also encourage the authors to consider and comment on my review points regarding a more in-depth and transparent discussion of the limitations and to reformulate wording in the paper accordingly (see my comments in 'Limitations' and 'Paper Formatting Concerns').

---

> ### Author Response · Authors · 2025-08-06
> **The missing section on "How Each Component Affects the Performance of Real-World Transfer"**
>
> **Dear Reviewer 6pVJ:**
>
> Thank you for your valuable feedback and for your interest in this work. We are encouraged to know that our responses have addressed some of your concerns.
>
> Regarding the missing section on "How Each Component Affects the Performance of Real-World Transfer," we apologize for the delay and are now able to provide a detailed numerical analysis. We have been working diligently to incorporate the valuable feedback from all reviewers.
>
> For this analysis, we sampled 100 prompts from both the in-distribution case (Forza-Horizon Scene) and the out-of-distribution case (Real-World Scene). We then generated 96-second videos using these prompts across stage 3 and stage 4, and 6-second videos across stage 1 and stage 2. To assess performance, we measured the FVD, moving-PSNR, and moving-LPIPS scores. These metrics help us understand how each component influences the generalization ability of the model. The results are provided below.
>
>
> | **Stage / Setting**                                 | **FVD ↓** | **Moving-PSNR ↑** | **Moving-LPIPS ↓** |
> |-----------------------------------------------------|-----------|-------------------|--------------------|
> | Warmup (In Distribution Cases)                      | 1518      | 8.6              | 0.79               |
> | Warmup (Out of Distribution Cases)                  | 3101      | 8.5               | 0.80               |
> | +Interactive Module (In Distribution Cases)         | 1398      | 10.5              | 0.64               |
> | +Interactive Module (Out of Distribution Cases)     | 2908      | 9.2               | 0.73               |
> | +Swin-DPM 6s (In Distribution Cases)                | 1331      | 10.4              | 0.67               |
> | +Swin-DPM 12s (In Distribution Cases)               | 1117      | 10.2              | 0.69               |
> | +Swin-DPM 24s (In Distribution Cases)               | 1107      | 10.0              | 0.71               |
> | +Swin-DPM 48s (In Distribution Cases)               | 1376      | 9.9               | 0.71               |
> | +Swin-DPM 96s (In Distribution Cases)               | 1227      | 10.0              | 0.71               |
> | +Swin-DPM 6s (Out of Distribution Cases)            | 2316      | 9.2               | 0.73               |
> | +Swin-DPM 12s (Out of Distribution Cases)           | 1811      | 9.0               | 0.74               |
> | +Swin-DPM 24s (Out of Distribution Cases)           | 1663      | 9.0               | 0.75               |
> | +Swin-DPM 48s (Out of Distribution Cases)           | 2169      | 8.9               | 0.76               |
> | +Swin-DPM 96s (Out of Distribution Cases)           | 1836      | 8.9               | 0.76               |
> | +SCM 6s (In Distribution Cases)                     | 1469      | 10.4              | 0.67               |
> | +SCM 12s (In Distribution Cases)                    | 1373      | 10.2              | 0.67               |
> | +SCM 24s (In Distribution Cases)                    | 1318      | 9.6               | 0.71               |
> | +SCM 48s (In Distribution Cases)                    | 1317      | 9.9               | 0.69               |
> | +SCM 96s (In Distribution Cases)                    | 1564      | 9.6               | 0.70               |
> | +SCM 6s (Out of Distribution Cases)                 | 2125      | 9.0               | 0.72               |
> | +SCM 12s (Out of Distribution Cases)                | 1541      | 9.1               | 0.72               |
> | +SCM 24s (Out of Distribution Cases)                | 1264      | 9.2               | 0.74               |
> | +SCM 48s (Out of Distribution Cases)                | 1515      | 9.0               | 0.74               |
> | +SCM 96s (Out of Distribution Cases)                | 1415      | 9.1               | 0.74               |

---

> ### Author Response · Authors · 2025-08-06
> **Concerns Regarding Limitations**
>
> Thank you for raising this. We will add a limitation discussion as follows:
>
> **Limitations**
>
> While **The Matrix** shows strong potential, there are several areas that remain challenging and should be the focus of future research and development:
>
> 1. **Physical Implausibilities and Temporal Inconsistencies**: While the model generates visually impressive videos, there are occasional physical implausibilities and temporal inconsistencies. These aspects highlight opportunities for enhancing realism in future iterations.
>
> 2. **Limited Training Scope**: Currently, **The Matrix** has been trained on only two labeled games (Forza Horizon and BMW). While these initial results show great promise, broadening the model’s training across a wider variety of games and environments will be crucial for improving generalization and real-world transferability.
>
> 3. **Handling of Dynamic Objects**: **The Matrix** performs well with static objects, such as the BMW car, but handling dynamic objects that exhibit complex behaviors remains an ongoing challenge. Addressing this limitation will be essential for expanding the model's capabilities in more dynamic, real-world scenarios.
>
> These limitations reflect important areas for future study and refinement. While **The Matrix** is a strong proof of concept, overcoming these challenges will enhance its applicability and performance in diverse environments and more complex scenarios.
>
> ---
>
> We sincerely hope that our response addresses all of your concerns. We would be truly encouraged to receive any additional suggestions for further improving this work. Please feel free to let us know if you have any further comments or recommendations!

---

### Official Review · Reviewer_7MCB · 2025-06-30

**Clarity:** 2
**Significance:** 4
**Originality:** 3
**Rating:** 4
**Confidence:** 3

**Summary:**

This paper proposes a novel video generation model for generating infinitely long videos efficiently with action control. It also proposes a new dataset consisting of labeled game data for training.

**Questions:**

1. Why does the interactive module convert action labels into text prompts instead of directly conditioning on the action for more precise control?
2. How is the unlabelled real-world/synthetic data from the source dataset constructed? Can the authors provide some examples of these unlabeled data and explain how they differ from the video data used to train the base video model?
3. Most qualitative results focus on the Forza Horizon variant of the model, with only movement control with "WASD". Can the authors provide more results trained on Cyberpunk 2077 and the DROID dataset on camera control/robot joint control? The 4th and 5th row in Fig. 5 are too vague to demonstrate the model's controlability with the Cybepunk data and the 6th row in in Fig 5. doesn't show the control space for the robot grasping data. Also, are these model variants also have strong generatibilty to unseen environments like the  Forza Horizon varaint?
4. Why does the action space of the Cybepunk data only encode forward motion and the four directions of the camera motion? Will it not work if both the 4 direction camera motion and the 4 direction agent motion are encoded?
5. Does the model support joint training on all the game data with a uniform action representation, rather than fine-tuning a different variant to accommodate the different action spaces of the different subsets?

**Ethical Concerns:**

["NO or VERY MINOR ethics concerns only"]

**Final Justification:**

After reading the author's rebuttal and the other reviewer's comments, I drop my rating to weak accept as I still find most of the technical contributions are engineering implementations without much research insights in video generation. Some of the claims also lack supportive quantitative numbers and video examples and I suggest the authors include them in the final manuscript. However, I'm still in favor of accepting this work as a strong open-sourced baseline for further research in world generation.

**Limitations:**

yes

**Paper Formatting Concerns:**

1. The paper has an embedded video, and it also doesn't work with my PDF reader. I'm wondering whether this is allowed in the formatting instructions.

**Quality:**

3

**Strengths And Weaknesses:**

Strengths:
1. The paper is easy to follow, and the infinite-long world generation task is well motivated with high application potential.
2. The proposed source dataset from video games is a promising solution for obtaining high-quality action-labeled video data. I appreciate the engineering effort of the authors to create such a dataset.
3. The qualitative results are strong and promising, showing the generalizability and various application aspects of the model.
4. The whole pipeline is sound with a good combination of existing modules and an extensive ablation study.

Weaknesses:
1. Some technical contributions are a bit overclaimed. The claimed high-quality rendering is the feature of the base video model, and the real-time generation is achieved with a stream consistency model, which is not the contribution of this work.
2. The current model still lacks some important features for world generation, such as realistic physics, interaction with other objects/characters, a memory mechanism for consistency, and precise movement/camera control. I would still call it a long-horizon video generation model with some vague controlability instead of a world model.
3. Missing technical details. 1). The warm-up stage in Fig. 2 is not explained in the method section. It seems from the caption that it is trained from "Synthesized Observations of Unreal Rendered Contextual Environments data", but later in the training details, the authors "warm up the base DiT model on unlabeled Source data", which is "real-world unlabelled footage". Which datasource is actually being used for warmup and how is the model trained in this step? 2). In Sec. 4.1, it says more details about the training datasets can be found in Appendix B.3, but this section is missing in the supplementary. I'm particularly curious about how the DROID robotics dataset is used in training.
4. The action space of the model is oversimplified. The authors map action to very high-level text prompts, sacrificing precise controllability.

---

> ### Author Rebuttal · Authors · 2025-07-31
>
> **Dear Reviewer 7MCB:**
>
> Thank you for your positive feedback and for appreciating our contributions to dataset development, pipeline design, and the qualitative results of our method. Below, we address your concerns in detail.
>
> ---
>
> ### **Concern 1: High-quality rendering as a feature of the base video model**
>
> We agree with your assessment that the base model contributes significantly to the rendering quality. However, we have made substantial efforts to ensure that the introduction of Swin-DPM, SCM, and control labels does not degrade visual quality. Generating long videos with stable visual quality remains a significant challenge in the video generation domain, and it is important to note that the base model alone does not possess this capability.
>
>
> ### **Concern 2: Real-time generation achieved by the consistency model**
>
> We apologize for any potential misunderstanding. While SCM utilizes the consistency model as its distillation method, this technique alone would not have achieved the real-time efficiency we observed.
>
> Although Swin-DPM does not improve inference speed at a global level (for instance, a 4-second video still takes approximately 4 minutes to produce), it dramatically reduces latency. Previously, you would have to wait 4 minutes just to see the first frame; now, the 4 minutes are evenly distributed across each video frame, meaning you can see 4 frames (1 second, containing 16 frames) every 25 seconds. This gives the consistency model a near real-time experience, whereas, without this optimization, you would have to wait for several seconds to view a short video.
>
> Additionally, we optimized the SCM framework significantly. In the initial setup, using just the consistency model + Swin-DPM would yield an inference speed of less than **4 fps**, due to several factors:
>
> - **GPU Utilization:** Swin-DPM does not fully utilize the GPU, preventing linear acceleration across multiple GPUs.
>
> - **Inefficient CPU Operations:** Certain operations within the DiT pipeline (e.g., calculating rope and other constants) are inefficient in real-time scenarios, limiting performance.
>
> - **NCCL and PCIe Communication Bandwidth:** In non-real-time scenarios, communication occurs at lower frequencies, so bandwidth limitations are less critical. However, in real-time settings, these limitations become significant.
>
> We have addressed these challenges, resulting in significant improvements to real-time inference speed. The code has been open-sourced for several months, and you are welcome to try it yourself. However, due to NeurIPS' rebuttal policy, we cannot provide direct links here, but you can search for it online.
>
>
> ### **Concern 3: Lack of important features for world generation**
>
> We acknowledge that Matrix is still an early exploration in the field of world models. However, we believe it demonstrates important capabilities that suggest the potential for future world-generation models. Specifically:
>
> - **Generalization Ability:** As reported in Section 4.3 and shown in Figures 7a and A3 in the appendix (if you experience issues viewing the appendix, we recommend using Chrome or Preview on Mac, as we will address this Adobe version conflict in the revision), Matrix allows for rich out-of-distribution generalization. This is due to its pretraining on a large amount of realistic unlabeled data, showing the potential for building general foundational world models through scaling up training and data size.
>
> - **Interaction with Objects:** Although we are not able to provide visual examples at this time, a small number of object interactions do exist in the Matrix. For example, when a car collides with a tree or brushwood, the tree shows breakage effects; when the car hits a building, it stops; and when transitioning from land to water, the interaction with the ground changes. While we acknowledge that this is not yet a fully generalized interaction across all objects, these examples suggest that the research trend is moving towards the capability of object interaction. We will add this discussion to the paper.
>
>
> ### **Concern 4: Oversimplification of the action space**
>
> We agree that simplifying the action space may sacrifice the precision of movement. However, this trade-off allows the model to generalize better to real-world settings. As explained earlier, generalization of control in out-of-distribution scenarios is achieved by altering text prompts. The model is pretrained from the DiT model and inherits its understanding of text-based commands. By randomly dropping control signals during training, the model learns to manipulate the movements of real-world objects.
>
> We hope to expand this further in future work by collecting additional types of interactions and using language as a unified protocol to represent them, enabling the model to create novel interactions. However, due to the high workload required, we did not implement this idea in the current work. This future direction also contributes to our decision to convert action labels into text.
>
> We further conducted an experiment to validate the effectiveness of using transfer action labels in the form of one-hot labels instead of text prompts. Due to time constraints during the rebuttal period, the model may not have been fully trained (note that the numerical results may not be aligned with the paper), but the preliminary results suggest that text prompt-style actions outperform one-hot labels across all metrics. Specifically, here User-Study Acc. suggesst we mannully look into generated videos and count how many percentage of frames we think is correctly aligned with controls.
>
> We hypothesize the following reasons for this result:
>
> - **Text Prompts Provide Semantically Rich Representations:** Text prompts offer distinct and semantically meaningful representations for actions. In contrast, one-hot labels are less expressive. For example, concepts like "press W for 1 second, then press S for 1 second" or "moving irregularly" are difficult to capture with one-hot labels but can be easily represented with text-based prompts.
>
> - **Base DiT Model Familiarity:** The base DiT model has been pretrained with text prompt embeddings, and it cooperates well with these embeddings. Since the model is already familiar with text-based representations, it is more effective when using them compared to one-hot labels.
>
>
> **Table: Comparison of Action Spaces on Evaluation Metrics**
>
> | Action Space | FVD ↓  | Moving-PSNR ↑ | Moving-LPIPS ↓ | User Study ACC ↑ |
> |--------------|--------|---------------|----------------|------------------|
> | one-hot      | 1311   | 10.42         | 0.590          | 0.72             |
> | text         | 1288   | 10.52         | 0.600           | 0.90             |
>
> ### **Concern 5: Construction of real-world data**
>
> Due to the rebuttal policy, we are unable to provide specific examples of the real-world data. However, we are happy to explain how this data is constructed and how it differs from video data used to train the base model. You can get an idea of what the data looks like by searching for keywords like "4K HUD walking in Tokyo, Shibuya Street." These videos share similar movement patterns to the game data, typically recorded from first- or third-person perspectives, with uniform movement speeds and high video quality. Unlike the base model, which uses a wide variety of videos from the internet, our real-world data focuses on walking, driving, and other movement scenarios captured from first- or third-person perspectives.
>
> ### **Concern 6: More cases in Cyberpunk and DROID and their generalization ability**
>
> While we are unable to provide additional video cases due to the rebuttal policy, we can address your concerns. In terms of generalization ability, the answer is yes for the Cyberpunk cases, but no for DROID. The DROID scenario is more specialized and narrow, making generalization to other domains more difficult. However, the Cyberpunk dataset generalizes well, as the base model is already familiar with urban scenes.
>
> ### **Concern 7: Why is the action space of Cyberpunk limited to four directions of camera movement and forward motion?**
>
> The main reason for limiting the action space is to reduce the complexity of the data. Encoding all possible walking directions and camera movements would result in over 18 control signals, increasing the amount of data required for training. This would place a heavy burden on data collection, filtering, and balancing. We experimented with this approach but abandoned it due to the extreme workload. While the current action space can logically produce similar movement effects to the 18 possible control signals, we decided to keep the model simple for this work.
>
> ### **Concern 8: Does the model support joint training on all game data with a uniform action representation?**
>
> Yes, the model does support joint training with a uniform action representation across different games. However, since we only collected data for two games, we did not implement this feature in the current work. The action space design, where all actions are first converted into text, was developed with this capability in mind, as previously explained.
>
> Below, we provide the training results for first training the Forza Horizon 5 scenario alone, followed by joint training on both datasets. The results demonstrate similar performance in terms of visual quality and movement fidelity.
>
> **Table: Evaluation Metrics for Joint Training on Different Scenarios**
>
> | Scenario                          | FVD ↓ | Moving-PSNR ↑ | Moving-LPIPS ↓ |
> |------------------------------------|-------|---------------|----------------|
> | Forza-Horizon 5                    | 1288  | 10.52         | 0.600          |
> | Forza-Horizon 5 + Cyberpunk 2077   | 1282  | 10.37         | 0.599          |
> ---
>
> We hope these clarifications address your concerns. Thank you once again for your thoughtful feedback and for considering our work.

---

> ### Comment · Reviewer_7MCB · 2025-08-01
>
> I appreciate the authors for their detailed rebuttal. I agree that text can act as a more uniform action condition signal than one-hot action vectors, and I thank the authors for sharing their results and insights. While the lack of method novelty and key technical contribution is still a weakness of this work, I believe the proposed whole pipeline for world generation, which consists of dataset curation, pretraining/fine-tuning, and technical details for high-quality and efficient rendering, outweighs the weakness and is valuable to the community.
>
> However, I find this project has only open-sourced the code for part of the whole pipeline. I hope the authors are being honest here and will release the code of the whole pipeline, including both data collection and model training in the future for better reproducibility, especially the data curation from the game engine, which could benefit many tasks besides video world generation.

---

> > ### Author Response · Authors · 2025-08-04
> > **Thank you for your support and valuble feedback!**
> >
> > **Dear Reviewer 7MCB:**
> >
> > Thank you very much for your support and valuable feedback! In response to your question about open-sourcing the entire pipeline, we plan to update the repository and share the link as soon as it is permitted by NeurIPS' official policies. This update will include the dataset curation process, the full dataset, model training procedures, and the real-time inference framework.

---

### Official Review · Reviewer_m9ww · 2025-07-01

**Clarity:** 4
**Significance:** 3
**Originality:** 4
**Rating:** 4
**Confidence:** 3

**Summary:**

This paper introduces "The Matrix," a world simulator that generates infinitely long, high-fidelity (720p) video streams with real-time, interactive control. The model is trained on a combination of supervised data from video games and unsupervised real-world footage. Key contributions include the GameData platform for automated data collection, the novel Swin-DPM technique for infinite video generation, and the demonstration of zero-shot generalization to new scenarios.

**Questions:**

Questions：
1. In line126, how much does the value of the probability q affect the performance of the model?

2. Have you tried different window stride besides s=1?

**Ethical Concerns:**

["NO or VERY MINOR ethics concerns only"]

**Final Justification:**

Based on the rebuttal and discussion, I will keep my score.

**Quality:**

3

**Strengths And Weaknesses:**

Strengths
1. The introduction of the GameData platform is a major strength. It addresses a critical bottleneck in training world models: the prohibitive cost and complexity of acquiring large-scale, action-labeled data.

2. The core technical contribution, the Shift-Window Denoising Process Model (Swin-DPM), presents a novel and well-reasoned approach to extending pre-trained diffusion models for infinite-horizon video generation.

3. The 1-minute video demo and other visual results are highly effective.

Weaknesses

1. The paper presents SCM as a key component in Figure 2, which could be interpreted as a novel part of the proposed architecture. However, the text clarifies that SCM is an adaptation of a prior method used to achieve real-time speeds. This is not a flaw in the method, but a point of potential confusion regarding the paper's specific innovations. I recommend that you clarify the role of SCM in Figure 2 and the main text. For example, you could relabel the component in the figure as "Acceleration via SCM" to explicitly attribute the technique and frame it as a crucial optimization step rather than a core generative model contribution. This would improve the clarity of the paper's originality.

2. The paper claims real-time generation (16 FPS), but this performance is contingent on the SCM. The core generative model, Swin-DPM, appears to have significant efficiency limitations due to its auto-regressive nature. Based on the evidence you provide in Table 1, The + Swin-DPM model runs at only 0.8 FPS. The jump to 16 FPS only occurs after applying SCM, which also degrades visual quality (FVD increases from 1651.50 to 1936.79). Please acknowledge that the core Swin-DPM model is not real-time by itself and that the real-time capability is achieved through a distillation process (SCM) that trades some visual fidelity for speed. This would provide a more complete and transparent assessment of the method's performance.

3. Hyperparameter Sensitivity (Line 126): During training, you replace labeled inputs with a default description with probability q=0.1. How sensitive is the model's control precision (e.g., Move-PSNR) to this value? Providing a brief ablation study or sensitivity analysis for q would help justify this choice and improve the work's reproducibility.

4. Stride Choice in Swin-DPM (Line 143): The paper states that Swin-DPM generates tokens with a stride of s=1. Have you experimented with larger strides (e.g., s > 1)? While a larger stride might risk temporal inconsistency, it could also improve the baseline efficiency of the model before SCM. A brief discussion on why s=1 was chosen would be valuable.

5. The manuscript would benefit from a thorough proofread to correct several small typographical errors (e.g., in the caption of Figure 2, "Swin-DPMenables" and "Modelis").

---

> ### Author Rebuttal · Authors · 2025-07-31
>
> **Dear Reviewer m9ww:**
>
> Thank you for your positive feedback. We are especially encouraged by your appreciation of the GameData platform, Swin-DPM, and the visual results of our methods. Below, we address your concerns in detail.
>
> ---
>
> ### **Concern 1: Contribution of SCM and Swin-DPM to real-time efficiency**
>
> We apologize for any potential misunderstanding regarding the contribution of SCM and Swin-DPM to real-time efficiency. While SCM uses consistency modeling as its distillation method, this technique alone would not be sufficient to achieve the final level of efficiency.
>
> Swin-DPM does not directly improve the inference speed of the videos from a global level (for example, a 4-second video still takes around 4 minutes to produce). However, it drastically decreases the latency. Previously, you would have had to wait 4 minutes to see just the first frame. Now, the 4 minutes are evenly distributed across each video frame, allowing you to see 4 frames (1 second of video, which contains 16 frames) every 25 seconds. This means the consistency model provides you with a near real-time experience, rather than the long wait that would be required to view several seconds of video in non-real-time settings.
>
> Furthermore, we have significantly optimized the SCM framework. In the initial setting, using only the consistency model and Swin-DPM would result in an inference speed of less than **4 fps**, due to the following reasons:
>
> - **GPU Utilization:** Swin-DPM does not fully utilize the GPU, which limits the parallel attention computation, preventing linear acceleration across multiple GPUs.
>
> - **Inefficient CPU Operations:** There are several inefficient CPU operations within the DiT pipeline (e.g., computing rope and other constants) that are typically not a concern in non-real-time scenarios but drastically limit real-time efficiency.
>
> - **NCCL and PCIe Communication Bandwidth:** In non-real-time scenarios, communication happens at a relatively low frequency, so bandwidth limitations are not significant. However, in our real-time setting, bandwidth becomes a critical issue.
>
> We have addressed all of these challenges, resulting in significant acceleration for real-time inference. All the code has been open-sourced for several months, and you can try it yourself if you're interested! However, due to NeurIPS' policy, we cannot provide direct links here. You can easily find the code by searching for it on Google.
>
>
> ### **Concern 2: Hyperparameter sensitivity**
>
> Thank you for raising this point! We conducted a numerical ablation study on the probability of the "replace label" (q), and the results are shown below. Additionally, the User-Study Accuracy is based on a manual inspection of the generated videos, where we counted how many frames we considered correctly aligned with the control inputs.
>
> **Table: Effect of Replace Rate on Evaluation Metrics**
>
> | Replace Rate      | FVD ↓ | Moving-PSNR ↑ | Moving-LPIPS ↓ | User Study ACC ↑ |
> |-------------------|-------|---------------|----------------|------------------|
> | 0.00              | 1344  | 10.42         | 0.600          | 0.89             |
> | 0.05              | 1314  | 10.45         | 0.594          | 0.87             |
> | 0.10              | 1311  | 10.44         | 0.594          | 0.88             |
> | 0.15              | 1318  | 10.44         | 0.594          | 0.86             |
>
> We believe the results demonstrate that while the hyperparameter is sensitive, we have identified an optimal range that provides the best trade-off between accuracy and robustness. We will include more details in the revised manuscript to clarify this.
>
> ### **Concern 3: Stride choices**
>
> Yes, we did experiment with larger stride sizes, and we observed that they can improve consistency and coherence in generation. However, this comes at the cost of increased computational burden. Larger strides result in larger feature sizes that need to be stored on the GPU, and the square rate of self-attention operations increases accordingly. Given these trade-offs, we chose a stride of 1 for the best efficiency, balancing computational load and performance. Attached is a numerical evaluation of stride size vs. video coherence using VBench (time cost increasing could be viewed as linear growth to the stride size). To collect those metrics, we select 100 prompts randomly sampled from the testing dataset, and generate 768 frames (48s) each from those prompts. We use the custom_input mode in VBench to evaluate those results under the above seven metrics.
>
>
> **Table: Effect of different strides to improve coherence (VBench Metrics)**
>
> | VBench Metrics         | Stride 1 | Stride 2 | Stride 3 | Stride 4 |
> |------------------------|----------|----------|----------|----------|
> | subject_consistency    | 0.862    | 0.873    | 0.882    | 0.885    |
> | background_consistency | 0.900    | 0.911    | 0.914    | 0.917    |
> | imaging_quality        | 0.662    | 0.675    | 0.689    | 0.695    |
> | motion_smoothness      | 0.976    | 0.976    | 0.978    | 0.977    |
> | dynamic_degree         | 0.99     | 1.00     | 1.00     | 1.00     |
> | aesthetic_quality      | 0.527    | 0.533    | 0.530    | 0.528    |
>
> ### **Concern 4: Improving manuscript writing and proofreading**
>
> Thank you for pointing this out! We will carefully polish the paper during the revision process to improve clarity and readability.
>
> ---
>
> We sincerely hope that these clarifications address your concerns. Thank you again for your thoughtful feedback and for considering our work.

---

> > ### Comment · Reviewer_m9ww · 2025-08-04
> > **Response to authors**
> >
> > Thank you for your detailed rebuttal and for providing the additional ablation studies in response to my questions. The new tables regarding the hyperparameter `q` and the stride choice `s` are particularly helpful and appreciated.
> >
> > However, after reviewing your response and the comments from other reviewers, my primary concern regarding the clarity and framing of the contributions remains. While your rebuttal clarifies the technical aspects, it does not fully resolve the core issue of how the paper's contributions are presented.
> >
> > As I and other reviewers (e.g., 7MCB, 6pVJ) have noted, there seems to be a disconnect between the core novel component (Swin-DPM) and the main claimed result (real-time generation). Your rebuttal confirms that the real-time performance is not an intrinsic property of Swin-DPM but is achieved through significant optimization and a separate distillation-like step (SCM), which comes with a quality trade-off. While the engineering effort is commendable, the current framing could be seen as overstating the capabilities of the core generative model itself.
> >
> > To help me reassess my score, I would appreciate clarification on the following points, which build upon my initial review and the broader discussion:
> >
> > 1.  **On the Choice of Action Space:** You argue that converting actions to text prompts aids generalization, and you provide a new experiment to support this. This is a compelling result. However, as Reviewer 7MCB pointed out, this also significantly simplifies the control problem. Could you elaborate on whether this text-based approach was a deliberate *a priori* design choice for its generalization properties, or a practical simplification to avoid the engineering complexity of conditioning on more direct, continuous control signals? Understanding this motivation is key to evaluating the novelty of the Interactive Module.
> >
> > 2.  **On Performance Transparency:** Thank you for the table showing how coherence metrics change with stride `s`. To make the trade-off fully transparent, could you augment this by providing the inference latency or FPS for each stride setting *before* the SCM is applied? This would directly address whether larger strides offer a meaningful native efficiency improvement for Swin-DPM, or if the system remains fundamentally reliant on SCM for real-time speeds across all stride settings. This information is critical for me to accurately judge the efficiency claims and the true performance contribution of Swin-DPM versus the SCM.

---

> ### Author Response · Authors · 2025-08-06
> **On the Choice of Action Space**
>
> **Dear Reviewer m9ww,**
>
> Thank you for your insightful question! We appreciate the opportunity to clarify this aspect of our work. There are indeed considerations from both perspectives here, and we hope to provide a clearer explanation.
>
> First, the simplification does not actually occur when converting the control signal from 'WASD' to text prompts. 'WASD' are simply one-hot labels, conveying no more information than the text prompts themselves. Both 'WASD' and the text prompts vary across frames (each frame has an individual, different prompt 'the care is driving xxx' to describe the action of the exact current frame), so from an informational perspective, they are essentially the same thing.
>
> The true simplification comes from the decision to adopt only 'WASD' as the action space. We do collect more detailed data, such as the car's speed along the x, y, and z axes, the angular velocity of the tires, and other state parameters. However, we ultimately chose not to use these values in the control for the following reasons:
>
> 1. **Causality and Simplicity**: The parameters we collect (e.g., speed, tire angles, accelerations) are outcomes of the 'WASD' controls. From the perspective of causal reasoning, 'WASD' provides the simplest and most direct signal. The main goal was to focus on the cause of movement, and 'WASD' serves as the clearest, most efficient representation of this cause. Whether we describe it as "pressing 'W'" or "the car moving forward," both convey the same information at a logical level.
>
> 2. **User Experience**: One of our main design goals is to provide a user-friendly interaction that mirrors traditional game controls. Asking users to provide detailed per-frame data such as speed or other state parameters would complicate the process and take away from the experience. Simplifying the action space to 'WASD' makes the system more accessible, allowing users to interact intuitively with the model. Requiring additional data would likely overwhelm users, and simplifying control to something familiar, like 'WASD', helps achieve a more enjoyable user experience.
>
> 3. **Game Limitations**: In the case of some games like *Cyberpunk*, we cannot access detailed movement data due to the game's anti-cheat mechanisms. As such, we are limited to tracking 'WASD' and mouse movements. Additionally, obtaining detailed movement statistics like speed and acceleration would require modifying the game, which is often technically challenging and potentially against the game’s terms of service. This is why we did not pursue this route in the current implementation.
>
> However, in other environments such as droid simulations, we do use precise angular movements as input (as shown in the appendix). In these cases, we chose not to convert those precise movements into text prompts because we could not find appropriate textual descriptions for the complex movements of all seven motors involved.
>
> ---
>
> We hope this explanation provides a clearer understanding of our approach. There are valid reasons for choosing 'WASD' as the control space, but we also recognize the trade-offs and the simplifications involved. If you have any further questions or suggestions, we would be happy to clarify further.

---

> > ### Author Response · Authors · 2025-08-06
> > **On Performance Transparency**
> >
> > **Dear Reviewer:**
> >
> > Thank you for raising this point! We appreciate the opportunity to clarify the differences in FPS across the various configurations.
> >
> > The combined table is as follows:
> >
> > | Metric / FPS            | Stride 1 | Stride 2 | Stride 3 | Stride 4 |
> > |-------------------------|----------|----------|----------|----------|
> > | subject_consistency     | 0.862    | 0.873    | 0.882    | 0.885    |
> > | background_consistency  | 0.900    | 0.911    | 0.914    | 0.917    |
> > | imaging_quality         | 0.662    | 0.675    | 0.689    | 0.695    |
> > | motion_smoothness       | 0.976    | 0.976    | 0.978    | 0.977    |
> > | dynamic_degree          | 0.990    | 1.000    | 1.000    | 1.000    |
> > | aesthetic_quality       | 0.527    | 0.533    | 0.530    | 0.528    |
> > | **FPS (4 cards)**       | 0.47     | 0.42     | 0.30     | 0.20     |
> > | **FPS (8 cards)**       | 0.80     | 0.75     | 0.55     | 0.47     |
> >
> > By the way,  if you also consider the situation of applying SCM, the difference in speed will become much more pronounced. Here's why:
> >
> > 1. **Pre-SCM**: Before SCMs, we require CFGs, meaning each GPU handles double the token length, leading to nearly 100% GPU utilization. This allows parallelization with multiple GPUs for flash attention computing, which can accelerate performance linearly as the number of GPUs increases.
> >
> > 2. **Post-SCM**: After applying SCMs, the stride of 1 allows us to achieve performance with much less than 100% GPU utilization and very low GPU memory usage. This creates additional headroom for parallelizing **VAE**, leading to better resource usage and efficiency. With this configuration, we can still achieve close to linear acceleration when combining **VAE** parallelization. However, in the case of using a larger stride, while you still don’t reach 100% GPU utilization, the memory usage increases significantly. In this scenario, simply parallelizing **DiT** across multiple GPUs does not result in a linear increase in speed. Linear speedup only occurs when GPU utilization is close to 100%, as seen with flash attention. Additionally, with SCMs, the issue is that when using a larger stride, there isn’t enough room to parallelize **VAE** effectively on top of it, both in terms of the remaining GPU usage and memory usage.
> >
> > 3. **Communication Efficiency**: The smaller stride also reduces the communication overhead of self-attention, which is perhaps the most important factor for non-NVLink machines like L40s or 4090, as it is well known that communication is significantly expensive in Nvidia GPU devices (definitely more precious than gold or Vibranium). By reducing this communication burden, we can achieve much better performance on these machines.
> >
> > ---
> >
> > We hope this explanation clarifies the relationship between the optimizations and the resulting FPS differences. If you have any further questions or need more details, feel free to reach out!

---

> > > ### Author Response · Authors · 2025-08-09
> > > **Eagerly Looking Forward to Your Reply!**
> > >
> > > **Dear Reviewer m9ww:**
> > >
> > > We are excited to hear your thoughts and truly hope that our response has addressed your concerns effectively! Your feedback is invaluable to us, and we are eager to hear any additional suggestions or ideas you may have. As the author-reviewer discussion period is drawing to a close, we want to make sure we make the most of your expertise to refine and improve this work further. Please feel free to share any additional comments or questions, as we are keen to continue enhancing the quality of our submission.
> > >
> > > If our previous reply has fully addressed your concerns, we would be thrilled and deeply grateful for your time and invaluable insights in helping us strengthen this work!
> > >
> > > Thank you so much for your thoughtful contributions!

---

### Official Review · Reviewer_jKm1 · 2025-07-01

**Clarity:** 4
**Significance:** 4
**Originality:** 3
**Rating:** 5
**Confidence:** 3

**Summary:**

*The Matrix* is a real-time world simulator that generates infinitely long, high-fidelity 720p video streams with frame-level control at 16 FPS. Trained on a mix of labeled game data and unlabeled real-world footage, it generalizes to unseen environments with zero-shot capability. Key innovations include the Swin-DPM for infinite video generation, an Interactive Module for user input integration, and SCM for real-time performance. A new dataset, Source, supports training with paired action-video data. The system demonstrates strong visual quality, precise control, and scalability, paving the way for AI-driven simulations in gaming, robotics, and virtual environments.

**Questions:**

- *The Matrix* shows impressive long-form generation (e.g., the 15-minute desert video). However, will the quality degrade over time in more complex scenarios like urban scenes? Quantitative evaluations would help assess long-term fidelity.
- Could you provide details on the human effort involved in data annotation—such as the number of annotators and total time spent? This would help assess the cost of transferring the system to new games.

**Ethical Concerns:**

["NO or VERY MINOR ethics concerns only"]

**Final Justification:**

I appreciate the authors' response and continue to hold a positive view of the submission.

**Quality:**

3

**Strengths And Weaknesses:**

**Strengths**

- Generates high-quality videos at real-time speed.
- Substantial effort is demonstrated, including platform development, data collection, and novel model design and training.
- The module design is intuitive and effective.
- The paper is well-written.

**Weaknesses**

- The core task is questionable. The system uses significant resources to generate immersive virtual worlds that could be more efficiently produced using traditional 3D game engines.
- The visual diversity of the generated worlds appears limited by the training data.
- There is little attention to world consistency or coherence—generated environments tend to be forgotten after just a few seconds.

---

> ### Author Rebuttal · Authors · 2025-07-31
>
> **Dear Reviewer jKm1:**
>
> Thank you so much for your positive feedback and for recognizing our contributions in systematic framework design, dataset collection, and data pipeline establishment, as well as the overall performance of our approach. Below, we address your concerns in detail.
>
> ---
>
> ### **Concern 1: Core task is generating virtual worlds, which could be easily produced using traditional 3D engines.**
>
> We appreciate your insightful assessment of the core task and the challenge of generalization in world models. Before addressing the generalization ability of our approach, we’d like to first defend the usage of world models from a few perspectives.
>
> - **Why world models are useful, even if they only do what traditional 3D engines can also do:**
>
> **Perspective 1: Scaling up 3D engines with data is difficult.**
> Imagine you are tasked with creating a sim2real simulation for a company that has built a new factory in a new location. To simulate this scenario, you might need to:
>
> 1. Collect numerous videos of the factory, as we did in the DROID dataset, and use them to train a world model, which requires minimal human labor.
>
> 2. Alternatively, you could recruit several CG engineers to collect 3D mesh data and develop an entire new 3D environment, which would require substantial time and debugging.
> While current 3D GS methods can help accelerate the second approach, they still require heavy intervention from experienced CG engineers, which is both inconvenient and expensive compared to the first method. While CG-based methods dominate today, we believe that world models can offer a more efficient, data-driven alternative for creating sim2real simulations.
>
> **Perspective 2: 3D engines cannot conveniently provide gradients for downstream tasks.**
> Although techniques like differentiable 3D rendering exist, traditional 3D game engines generally struggle to provide gradient information for downstream tasks. As a result, neural network tasks often require methods like reinforcement learning or Q-learning to handle the absence of gradients. World models, however, can provide gradients directly, making them much easier to use in such scenarios. We discuss this aspect in detail in Section A of the supplementary materials.
>
> - **Generalization ability of our method (differentiating it from traditional 3D engines):**
> As reported in Section 4.3 and shown in Figures 7a and A3 in the appendix (if you encounter any issues viewing the appendix, we suggest using Chrome or Preview on Mac, as we will address this Adobe version conflict in the revision), our model demonstrates strong out-of-distribution generalization. By pretraining on a large amount of realistic unlabeled data, our model can generalize to unseen scenarios, which is a key advantage over traditional 3D game engines.
>
>
> ### **Concern 2: Visual diversity of the generated world.**
>
> For the visual diversity aspect, we refer you to our earlier response regarding the generalization ability of our model. The diversity is a direct result of the pretraining on a wide range of real-world data, which enables our model to generate highly varied virtual worlds.
>
>
> ### **Concern 3: Generated world coherence over time.**
>
> Thank you for raising this critical point! Currently, we sacrifice long-term coherence for faster rendering speed. Specifically, we removed long-duration self-attention and use a small history cache. By increasing the range of the self-attention module or expanding the history cache, we can significantly improve consistency, though this will come at the cost of slower processing speed. Currently, we set a stride parameter for the Swin-DPM to trade off between speed and coherence. Stride=1 is used for scenarios pursuing extreme real-time experience ,while larger strides take care of coherence over longer time. We’ve conducted numerical experiments to validate this trade-off, and the results show that increasing the attention range through increasing the Swin-DPM stride size does indeed improve coherence.
>
> **Table: Effect of different strides to improve coherence (VBench Metrics)**
>
> | VBench Metrics         | Stride 1 | Stride 2 | Stride 3 | Stride 4 |
> |------------------------|----------|----------|----------|----------|
> | subject_consistency    | 0.862    | 0.873    | 0.882    | 0.885    |
> | background_consistency | 0.900    | 0.911    | 0.914    | 0.917    |
> | imaging_quality        | 0.662    | 0.675    | 0.689    | 0.695    |
> | motion_smoothness      | 0.976    | 0.976    | 0.978    | 0.977    |
> | dynamic_degree         | 0.99     | 1.00     | 1.00     | 1.00     |
> | aesthetic_quality      | 0.527    | 0.533    | 0.530    | 0.528    |
>
> To collect those metrics, we select 100 prompts randomly sampled from the testing dataset, and generate 768 frames (48s) each from those prompts. We use the custom_input mode in VBench to evaluate those results under the above seven metrics.
>
> ### **Concern 4: Will the quality degrade over time?**
>
> To evaluate long-term performance, we measured the FVD score over time, and the results are provided below. Our model demonstrates consistent and stable performance across long durations, with no significant degradation in quality. Although the first 4 seconds show substantial improvement, the quality remains stable throughout the remaining duration, with no noticeable decline.
>
> **Table: Evaluation Metrics Across Different Time Durations of The-Matrix**
>
> | Time Duration | FVD ↓ | Moving-PSNR ↑ | Moving-LPIPS ↓ |
> |---------------|-------|--------|---------|
> | 0-4s          | 1448  | 10.6   | 0.61    |
> | 4-8s          | 1952  | 10.0   | 0.64    |
> | 8-12s         | 2016  | 9.8    | 0.66    |
> | 12-16s        | 2070  | 9.4    | 0.67    |
> | 0-10s         | 1232  | 10.2   | 0.63    |
> | 10-20s        | 1952  | 10.0   | 0.64    |
> | 20-30s        | 2016  | 9.8    | 0.66    |
> | 30-40s        | 2070  | 9.4    | 0.67    |
> | 40-50s        | 2016  | 9.8    | 0.66    |
> | 50-60s        | 2070  | 9.4    | 0.67    |
>
> We also further evaluate the performance over time using the **imaging quality**metric from the VBench. We sample 100 prompts from the in-distribution case (Forza-Horizon Scene) and out-of-distribution case (Real-World Scene), and generate videos of 96 seconds with them. We then measure the **imaging quality** metric of those videos at different times as follows:
>
> **Table: Performance across different time windows and environments**
>
> |   Time             | 6s | 12s | 24s | 48s | 96s |
> |----------------|---------|----------|----------|-----------|-----------|
> | OOD Cases         | 0.676   | 0.645    | 0.606    | 0.595     | 0.583     |
> | In-Distribution  Cases        | 0.640   | 0.619    | 0.607    | 0.609     | 0.603     |
>
> While OOD cases witness a slight degradation of quality, in-distribution cases are much stable and barely lose quality as time grows. Both of the cases can still maintain no-collapse visual quality as time grows.
>
> ### **Concern 5: Human labor used in data annotation.**
>
> Almost the entire data collection process, except for the Cyberpunk dataset, was automated, as detailed in Section B of the appendix. However, some limited human labor was involved to maintain the realism of the training data.
>
> - For non-Cyberpunk data (e.g., DROID), we employed 10 annotators to review video clips and ensure they contained valid motion scenes (e.g., walking, running, juggling in the wild). This process took approximately 10 days to complete.
>
> - For Cyberpunk 2077 data, we recruited 20 human annotators to play the game and collect data. The collection process lasted 24 days (6 hours per day), during which approximately 2,880 hours of data was collected. This data was split into around 1.8 million 6-second video clips paired with control signals. We developed a visualization tool to assist annotators in making their keyboard inputs more uniformly distributed and used algorithms to filter out low-quality data (e.g., data where the player was stuck or made incorrect actions). The final dataset consisted of about 1 million valid data pairs.
>
> We hope this clarifies the data annotation process. Please let us know if you need further information.
>
> ---
>
> We sincerely appreciate your thoughtful feedback and hope that our revisions address your concerns. Thank you again for your consideration.

---

### Official Review · Reviewer_TvGJ · 2025-07-03

**Clarity:** 3
**Significance:** 3
**Originality:** 3
**Rating:** 5
**Confidence:** 3

**Summary:**

The authors present a diffusion-based interactive video generation / simulation model (one in which the generated visuals reflect user text prompts and respond to user game inputs) while incorporating a novel “Shift-Window Denoising Process Model (Swin-DPM)” to enable infinite length video generation. They also introduce the “GameData” platform (used to generate the “Source” dataset) which captures paired game state and video frames. The authors boast that their method has improved rendering resolution and speeds versus existing methods (lines 58-64) though the experimental quantitative proof of this is somewhat lacking.

The Matrix diffusion model pipeline is represented in Figure 2 and is composed of:

1) A pre-trained Video DiT backbone [14] (which the authors note in section 4.3 makes their method more robust to out-of-distribution scenarios),
2) An Interactive Module responsible for responsible for “translating keyboard inputs into natural language that guides video generation” (described on page 4),
3) A novel “Shift-Window Denoising Process Model (Swin-DPM)” (described on page 5 and Figure 4) where a shifting time sequence of video tokens is denoised simultaneously. After each token starting the queue is fully denoised the time-series advances with a new noisy input being added.
4) A Stream Consistency Model (based on [27]) used to accelerate denoising although is sparingly described aside from a reference to existing work being given.

The GameData platform to autonomously collect data of paired video and game state is described on lines 172-181 where the authors use a number of open-source tools to capture game state, remove user interface (UI) elements, and capture paired video frames.

__Novelty and Contributions:__
- Diffusion interactive video generation model with improved visual fidelity and rendering speeds versus existing methods.
- Propose a novel "Shift-Window Denoising Process Model (Swin-DPM)" for infinite video generation.
- Describe their GameData platform for autonomous collection of training data.

__Experiments:__

Experiments display qualitative results in terms of long-form video generation with movement controls (Figure 6), generation of unlabeled scenes with movement control (Figure 7 a), and pure video generation without user interactivity (Figure 7 b). Quantitative results for an ablation study of The Matrix’s components are shown in Table 1 where the authors note a general trade-off between rendering speed, rendering quality and control precision.

**Questions:**

- Can the authors elaborate on why retraining an existing baseline method with their dataset may not be feasible (lines 183-184) to provide a more quantitative comparison to benchmarks?

**Ethical Concerns:**

["NO or VERY MINOR ethics concerns only"]

**Final Justification:**

Following the authors' rebuttal, I have increased my score to accept. Most of my criticisms have been addressed in the rebuttal: additional comparison to Diamond baseline and further elaboration into runtime compute efficiency of model. Although, in the future it would be good to have a more extensive comparison to Diamond (or other method) in the multiple environments presented in this paper. Please see my comments to the author's rebuttal and original strengths and weaknesses section of my review for further details.

**Limitations:**

Yes. Limitations are listed by the authors on lines 264-268.

**Quality:**

2

**Strengths And Weaknesses:**

__Strengths:__

- __State of the art (significance):__ The authors boast state of the art virtual world generation accommodating textual prompts and user inputs which the authors assert possess greater visual fidelity and frame rate compared to previous methods. They also assert that their method has better long term video generation capabilities. However, it would have been nice to see a more quantitative comparison to baselines (see weaknesses section).
- __Novel Method (originality):__ Approach using diffusion models with a novel Shift-Window Denoising Process Model (Swin-DPM) to facilitate infinite video generation.
- __Manuscript Clarity:__ I feel that clarity, significance and originality were well done though experimental quality related to benchmarks could be somewhat improved (see weaknesses below).

__Weaknesses:__

- __Quantitative experimental analysis (quality):__ The authors note that “Since existing world models are not trained on Forza Horizon 5 or Cyberpunk 2077, it is unfair to compare The Matrix against prior world models both qualitatively and quantitatively.” (lines 183-184). Is it not possible to retrain one/some of these models on your data to obtain a more quantitative comparison to baselines? I currently feel that this work is weakened without a substantive comparison to baselines (experimental quality could be improved). The authors claim advances in rendering speeds and resolution compared to existing methods, though some quantitative experimental results to support this claim would of been appreciated.

- While the method can supposedly generate “infinite-length videos”, I would of found it insightful if there were some metrics to quantify/show the degradation (or lack of) of quality between the start to end frames of the generated video sequence.

- __Computational limitations (significance):__ As the authors note in the Limitations section (lines 264-268), the model currently runs on 8xA100 GPUs which makes its usage prohibitive.

__Current Assessment and Suggestions for Improvement:__

I have currently marked the work as borderline reject. I feel that the submission currently relies too heavily on qualitative comparisons to existing visual generation methods instead of quantitative testing. The authors assert that quantitative testing versus existing methods is not possible due to these models being trained on different data. Can further justification be given by the authors as to why retraining with the existing dataset is not possible (see questions section below).

__Minor Points:__
- In case these have not yet been resolved, there are several typos related to missing spaces: 1) In Figure 1: “Matrixis”, 2) In Figure 2: “Modulefor” and “Modelis”, 3) on line 221 “Modulesignificantly”

- I find the wording on line 65 to be somewhat odd “While we do not claim the Matrix to be a foundational world model” when the approach is described as “foundational” several times earlier (line 1, Figure 1, line 46).

- My PDF viewer (Adobe Acrobat version 2025.001.20531) failed to be able to view the embedded videos and reported the error “A 3D data parsing error has occurred.”

---

> ### Author Rebuttal · Authors · 2025-07-31
>
> **Dear Reviewer TvGJ:**
>
> Thank you for your valuable feedback and suggestions! We greatly appreciate your recognition of the novelty, significance, and originality of our work. Below, we address your concerns in detail.
>
> ---
>
> ### **Concern 1: Why is quantitative analysis against previous methods difficult?**
>
> We understand your point about the importance of including quantitative comparisons with previous methods. While we agree that such comparisons would strengthen the paper, we faced several challenges that prevented us from including them in the main paper. However, during this rebuttal period, we have conducted a comparison with Diamond. We hope this addition will help clarify the challenges we encountered.
>
> - **Reason 1: Limited Open-Source Options:**
>
> Among the competitors listed in Table 2, only Diamond and Oasis offer limited open-source code, both of which were primarily trained on the Minecraft dataset. Other methods do not provide their code or training pipeline, making it difficult to reproduce their results, even using their own datasets.
>
> - **Reason 2: Dataset and Architecture Differences:**
>
> Another challenge we encountered is the significant difference between the Minecraft dataset (used by Diamond and Oasis) and the datasets we used, such as Forza Horizon and Cyberpunk. This mismatch in data labeling and network architecture made it challenging to apply their models directly to our data without significant modifications. This is why we opted not to include a comparison in the main paper.
>
> Finally, we conducted a comparison between our method and the Diamond world model in the Forza Horizon 5 setting. To ensure a fair comparison, both models were trained from scratch using 50,000 data clips over 4 full epochs. We used the default CSGO settings for Diamond. While Diamond requires additional labels like mouse clicks, weapons, etc., we randomly selected one example from the Diamond training set and inherited all training labels from it, except for the WASD direction label. While this approach might introduce some unintended biases, it was the most practical choice given the constraints.
>
> The results are provided below. Please note that the results may not align exactly with the ones reported in the original paper. Additionally, the User-Study Accuracy is based on a manual inspection of the generated videos, where we counted how many frames we considered correctly aligned with the control inputs.
>
> **Tab. Comparason with Diamond on Forza-Horizon 5 Dataset**
> | **Method**          | **FVD ↓** | **Moving-PSNR ↑** | **Moving-LPIPS ↓** | **User-Study Acc ↑** |
> |-----------------------------|-----------|------------|-------------|-----------|
> | Diamond                    | 2111      | 9.44      | 0.703       | 0.51      |
> | The-Matrix                   | 1314      | 10.45      | 0.588       | 0.92      |
>
> ### **Concern 2: Degradation of visual quality vs. time**
>
> Thank you for your insightful suggestion! We are pleased to provide an FVD vs. time table below to demonstrate the effectiveness of our method in generating long-duration videos. While the initial 4 seconds show significant improvement, the remaining time durations do not exhibit noticeable degradation in quality.
>
> **Table: Evaluation Metrics Across Different Time Durations of The-Matrix**
>
> | Time Duration | FVD ↓ | Moving-PSNR ↑ | Moving-LPIPS ↓ |
> |---------------|-------|--------|---------|
> | 0-4s          | 1448  | 10.6   | 0.61    |
> | 4-8s          | 1952  | 10.0   | 0.64    |
> | 8-12s         | 2016  | 9.8    | 0.66    |
> | 12-16s        | 2070  | 9.4    | 0.67    |
> | 0-10s         | 1232  | 10.2   | 0.63    |
> | 10-20s        | 1952  | 10.0   | 0.64    |
> | 20-30s        | 2016  | 9.8    | 0.66    |
> | 30-40s        | 2070  | 9.4    | 0.67    |
> | 40-50s        | 2016  | 9.8    | 0.66    |
> | 50-60s        | 2070  | 9.4    | 0.67    |
>
> We also further evaluate the performance over time using the **imaging quality**metric from the VBench. We sample 100 prompts from the in-distribution case (Forza-Horizon Scene) and out-of-distribution case (Real-World Scene), and generate videos of 96 seconds with them. We then measure the **imaging quality** metric of those videos at different times as follows:
>
> **Table: Performance across different time windows and environments**
>
> |   Time             | 6s | 12s | 24s | 48s | 96s |
> |----------------|---------|----------|----------|-----------|-----------|
> | OOD Cases         | 0.676   | 0.645    | 0.606    | 0.595     | 0.583     |
> | In-Distribution  Cases        | 0.640   | 0.619    | 0.607    | 0.609     | 0.603     |
>
> While OOD cases witness a slight degradation of quality, in-distribution cases are much stable and barely lose quality as time grows. Both of the cases can still maintain no-collapse visual quality as time grows.
>
>
> ### **Concern 3: High computation cost (8 A100 GPUs)**
>
> We are glad to report that our method now runs smoothly on 8 or even 4 L40 GPUs, utilizing a PCIe 4.0 connection with 64GB/s throughput—no need for NCCL. The rendering speed is over 32 FPS, which significantly reduces the computational cost to approximately $0.5 per hour for users. This represents a giant leap in both cost and speed compared to other world models based on diffusion methods!
>
> Furthermore, the entire codebase (which utilizes VNC to broadcast video frames from a remote server) has been open-sourced on GitHub for several months. It has already supported various startups working on world model products. The documentation includes detailed tutorials on how to use the code, and it is easy to set up. Unfortunately, due to NeurIPS policy restrictions, we cannot provide direct links to the code and supported products in this rebuttal, but we encourage you to explore these resources if you're interested. Attached is the GPU running time report of the Matrix DiT backbone under different settings.
>
> | GPU      | GPU Numbers | Communication | Latency for 1 Video Token | FPS |
> |----------|-------------|---------------|--------------------------|-----|
> | RTX4090 (L40s)  | 8           | PCIE 4.0      | 0.127s                   | 32  |
> | RTX4090 (L40s)  | 4           | PCIE 4.0      | 0.220s                   | 18  |
> | A100     | 8           | NVLink 4      | 0.101s                   | 40  |
> | A100     | 4           | NVLink 4      | 0.133s                   | 30  |
> | H100     | 8           | NVLink 4      | 0.077s                   | 51  |
>
>
> ### **Concern 4: Typos**
>
> Thank you for your careful proofreading! We will address these typos in the revised version of the paper.
>
> ### **Concern 5: Use of the word "foundational"**
>
> Thank you for pointing that out! We will replace "foundational" with "novel" to improve the coherence of the context.
>
> ### **Concern 6: Document Viewing Issues**
>
> We apologize for the inconvenience you experienced. This may have been due to a version conflict in Adobe Acrobat. We will ensure this issue is resolved in the revision.
>
>
> ---
> We sincerely hope these clarifications and additions address your concerns. Thank you again for your thoughtful feedback, and we appreciate your consideration of our work.

---

> ### Comment · Reviewer_TvGJ · 2025-08-03
>
> Thank you for responding to my review and addressing my criticisms. I appreciate the additional Diamond baseline and improvements additional testing on various GPU setups to illustrate model efficiency in terms of runtime compute (though this was a more minor criticism). Although, in the future it would be good to have a more extensive comparison to Diamond (or other method) in the multiple environments presented in this paper. I have increased my score to reflect the authors' feedback.

---

> > ### Author Response · Authors · 2025-08-04
> > **Thank you for your support and valuble comments!**
> >
> > **Dear Reviewer TvGJ:**
> >
> > We are deeply grateful for your support and thoughtful feedback on our work. Your insights are invaluable and will undoubtedly help us improve and enhance the quality of this research. Thank you sincerely for your time and effort!

---

### Comment · Area_Chair_jvRG · 2025-08-02
**Please do keep the review process blind**

Dear Reviewers,

The authors have mentioned their code is open source and available in GitHub and encourage you to search for it if you are interested. Please do refrain to do so to keep the review process confidential, even at the cost of not being able to judge the code itself. Please do base your comments and recommendation on the submitted paper and supplementary material as well as in the discussion during the rebuttal period.

Also, note the although the authors wanted to provide additional videos to address comments about qualitative results, I explicitly asked the not to do so due to NeurIPS policy. So, please do understand those comments cannot be addressed and make your recommendation based on the evidence they have provided in their submission and during this rebuttal period.

Thanks
AC

---

### Decision · Program_Chairs · 2025-09-17

**Decision:**

Accept (poster)

**Comment:**

The paper introduces a video generation model capable of producing long, high-fidelity 720p video streams with interactive control at 18-50 fps. The model relies on a novel diffusion technique, coined Shift-Window Denoising Process (Swin-DPM) to generate long high-quality videos. In addition, developing the model required building a video game data collection tool, with which the authors created a dataset from Forza Horizon 5 and Cyberpunk 2077, which the authors plan to release.

The reviewers agreed on the impressive quality and length of the generated videos and appreciated the engineering effort, including data collection, and the commitment to open-source data and parts of their code. But they also had some concerns about lack of evidence of some claims, especially that Swin-DPM might not be key for fast image generation, lack of comparison to baselines, a simplified action space, quality degradation with time, or missing technical details.
The authors put effort in the rebuttal, performing more experiments, more notably the comparison with Diamond (a relevant baseline) and ablations to understand the impact of Swin-DPM and SCM on inference time, and addressed most of the concerns, but not all. Still all reviewers agreed the merits outperform the limitations and support acceptance.

I agree with the reviewers on the significance, engineering depth, and potential impact as a strong baselines for future research in world models. However, my recommendation is conditional on two factors:
1. The authors must improve the final version with the insights from the discussion during the rebuttal period, especially around the controversy about the claim that Swin-DPM is the key element that contributes to real-time performance as opposed to the optimization of SCM.
2. The chairs agree there are not ethical concerns on the data collection (see my comments to SAC below).

Regarding my confidence level:
* The controversy about the role of Swin-DPM makes me not recommend spotlight, but if the authors address it properly in the final version this could be bumped up.
* The ethical concern is the main reason for my confidence level on whether the paper should be bumped down to reject.